# Smooth Probabilistic Interpolation Benefits Generative Modeling for Discrete Graphs

## Abstract

Though typically represented by the discrete node and edge attributes, the graph topological information can be sufficiently captured by the graph spectrum in a continuous space. It is believed that incorporating the continuity of graph topological information into the generative process design could establish a superior paradigm for graph generative modeling. Motivated by such prior and recent advancements in the generative paradigm, we propose Graph Bayesian Flow Networks (GraphBFN) in this paper, a principled generative framework that designs an alternative generative process emphasizing the dynamics of topological information. Unlike recent discrete-diffusion-based methods, GraphBFN employs the continuous counts derived from sampling infinite times from a categorical distribution as latent to facilitate a smooth decomposition of topological information, demonstrating enhanced effectiveness. To effectively realize the concept, we further develop an advanced sampling strategy and new time-scheduling techniques to overcome practical barriers and boost performance. Through extensive experimental validation on both generic graph and molecular graph generation tasks, GraphBFN could consistently achieve superior or competitive performance with significantly higher training and sampling efficiency.

## 1 Introduction

Generative modeling of graph-structured data is an important task with applications in various important scenarios, *e.g.* molecule generation (Jin et al., 2018; Zang & Wang, 2020), traffic modeling (Yu & Gu, 2019) and protein design (Ingraham et al., 2019). With the development of deep generative models, there have been fruitful lines of research conducted to tackle the challenges, *e.g.*, Generative Adversarial Networks (Martinkus et al., 2022; De Cao & Kipf, 2018), Variational Auto-encoders (Jin et al., 2018; 2020), and Autoregressive models (You et al., 2018).

Recently, diffusion models (DMs) (Song & Ermon, 2019; Ho et al., 2020) have emerged as a powerful generative model across various fields (Zeng et al., 2022; Li et al., 2022). Learning DMs for graphs is challenging due to the discrete and complex relational nature of graphs. Niu et al. (2020); Jo et al. (2022) propose to dequantize the node vector and adjacent matrix with uniform/Gaussian noise and learn Gaussian DMs over the dequantized variables. However, these approaches face problems due to the incompatibility of continuous Gaussian diffusion processes with discrete data. For instance, they involve noisy graph samples that lack well-defined topological information, *e.g.*, clusterability or connectivity. To alleviate these problems, Vignac et al. (2022) proposed discrete graph diffusion models based on discrete DMs (Austin et al., 2021). It utilizes a structured categorical corruption process (corresponding to successive graph edit operations such as additions, deletions, and replacements) to destroy the graph and learns to generate by reverting this process. Such discrete formulation ensures noisy samples are valid discrete graphs with well-defined graph topologies. This enables the discrete DMs to extract rich graph spectral features, such as graph spectrums, that aid the generation process, thus achieving better performance.

However, though always ensuring valid graph topology, the discrete graph diffusion process is very unbalanced. Some graph editing steps can significantly change the adjacency matrix and unexpectedly perturb the graph topology, making it difficult to learn a satisfactory model for the reverse diffusion process. Recently, utilizing certain kinds of topological information formed in continuous space to provide smooth and consistent information flow for the generative process has been demonstrated

to be effective for graph generative model (Martinkus et al., 2022; Vignac et al., 2022; Jo et al., 2024). Based on such phenomena, we hold the intuition that stabilized dynamics of graph topological information during the generation process could be favorable for discrete graphs. Furthermore, we aim to design a new generative framework for discrete graphs that could conduct a smooth transition from data to noise, where the smoothness is quantified by generalized graph topological features.

Well motivated by this philosophy, we bring the concept of Bayesian Flow Networks, a recent advancement of generative paradigms, into the context of graph generation. Different from the discrete diffusion model whose latent variable is the discrete variable following a predefined forward categorical noisy distribution, the latent variable of the Bayesian Flow Network could be defined with the counts of discrete variables by sampling *infinite times* from the same forward categorical noisy distribution and is hence continuous. By projecting the continuous latent of BFN into probability simplex, we obtain a mixed-state representation of different discrete states which enables the smooth interpolation between an uninformative prior to the concrete observation. Based on such appealing property, we propose a new and general graph generative framework named Graph Bayesian Flow Networks (GraphBFN). By substituting the discrete variable to prob-

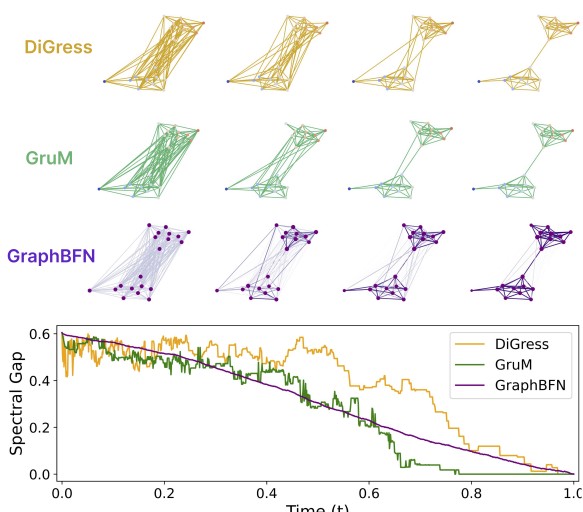

Figure 1: The top figures compares the sampling trajectories of GraphBFN to DiGress and GruM (Vignac et al., 2022; Jo et al., 2024). A smaller spectral gap reflects better graph clusterability (i.e. clearer sample). Both the visualizations and curves demonstrate that GraphBFN allows a much smoother transformation of graph topology information.

ability simplex in the intermediate state of the generative process, GraphBFN makes a probabilistic generalization of the adjacency matrix as *probabilistic adjacency matrix* whose entries represent the probability of corresponding edge type to emerge. Fortunately, graph topological features, like the graph spectrum, easily generalize to the *probabilistic adjacency matrix*, enabling straightforward analysis and the implementation of effective techniques such as spectrum conditioning. Quantitative measurements in Fig. 1 show that GraphBFN enables stabilized topological dynamics compared to diffusion-based methods with the target graph topology being smoothly recovered during the generation process. Specifically, our contribution can be summarized as follows:

- We firstly prove the concept of smooth transformation over topological information could facilitate the discrete graph generation with a new method termed GraphBFN based on Bayesian Flow Networks. We empirically demonstrated the effectiveness of smooth topological recovery. GraphBFN could fit more general graphs with rich node and edge features.

- To successfully implement the method and boost empirical performance, we introduce several innovations to sampling and training, including the adaptive sampling strategy which provides better tradeoffs between quality and diversity, a general time-scheduling that shows good compatibility with low entropy prior, and a novel graph spectrum conditioned strategy for large graphs.

- We conduct extensive experiments and ablation studies on several benchmarks including both abstract and 2D molecules. The empirical results show that the GraphBFN can consistently achieve superior or competitive performance in generating realistic and diverse graphs. Furthermore, GraphBFN also enjoys a high sampling efficiency with up to $10\times$ speedup while matching the performance diffusion-based methods.

## 2 PRELIMINARY

### 2.1 NOTATIONS AND BACKGROUND

We define the space of undirected graphs as $\mathcal{G} = (\mathcal{V}, \mathcal{E})$, where $\mathcal{E}$ represents a categorical edge space and $\mathcal{V}$ a node space. For a graph $\mathbf{G}$ with $n$ nodes, the adjacency matrix $\mathbf{E} \in \mathbb{R}^{n \times n \times c_e}$ and the node vector $\mathbf{V} \in \mathbb{R}^{n \times c_n}$ are both represented using one-hot encoding. Here, $c_e$ and $c_n$ denote the respective cardinalities. The absence of an edge is treated as a specific edge type. This framework encompasses both abstract graphs and 2D molecules studied in this work.

Then, we introduce the basics of spectral graph theory for better illustration. Spectral graph theory explores the relationships between graph properties and the eigenvalues of associated matrices, such as the adjacency matrix or Laplacian matrix. Previous works (Martinkus et al., 2022; Vignac et al., 2022) have demonstrated the effectiveness of conditioning on graph spectra in generating graphs. This motivates us to revisit the generative process from the perspective of building graph spectra, as shown in Fig. 1. Detailed discussions on spectral graph theory are presented in Appendix B.

### 2.2 BAYESIAN FLOW NETWORKS

In this section, we aim to provide a very intuitive introduction to the Bayesian Flow Networks (BFNs) (Graves et al., 2023) and leave an in-depth discussion and more details in Appendix A. Similar to the diffusion models, the BFNs also take the form of latent variable models with a series of noisy samples $\mathbf{y}_{\{1:n\}} = \langle \mathbf{y}_1, \cdots, \mathbf{y}_n \rangle$ of observation $\mathbf{x}$ are introduced as latent. We note that $\mathbf{y}_1$ of the start of the sequence refers to the uninformative sample of pure noise, which is the opposite of the diffusion literatures. Besides, BFNs also optimize the *variational lower bound* of the likelihood:

$$\log p_\phi(\mathbf{x}) \geq \mathbb{E}_{\mathbf{y}_{\{1:n\}} \sim q} \left[ \log \frac{p_\phi(\mathbf{x}|\mathbf{y}_{\{1:n\}}) p_\phi(\mathbf{y}_{\{1:n\}})}{q(\mathbf{y}_{\{1:n\}}|\mathbf{x})} \right] = -D_{KL}(q \| p_\phi(\mathbf{y}_{\{1:n\}})) + \mathbb{E}_{\mathbf{y}_{\{1:n\}} \sim q} \log \left[ p_\phi(\mathbf{x} \mid \mathbf{y}_{\{1:n\}}) \right] \tag{1}$$

where $q$ is a predefined distribution for creating the noisy samples similar to the noise-application distribution in the forward process of diffusion models (sender distribution) and $p_\phi$ refers to the probabilistic distribution(receiver distribution) implied by BFNs. From the graphical model perspective, BFNs differs from diffusion models in the construction of latent variables $\mathbf{y}_{\{1:n\}} = \langle \mathbf{y}_1, \cdots, \mathbf{y}_n \rangle$, which is defined as non-Markovian and could be extended as:

$$p_\phi(\mathbf{y}_1, \ldots, \mathbf{y}_n) = p_\phi(\mathbf{y}_1) \prod_{t=2}^{n} p_\phi(\mathbf{y}_i \mid \mathbf{y}_{\{1:t-1\}}) \tag{2}$$

The $p_\phi(\mathbf{y}_i \mid \mathbf{y}_{\{1:t-1\}})$ could be further represented with a predefined deterministic aggregation function of the process as $p_\phi(\mathbf{y}_i \mid f(\mathbf{y}_{\{1:t-1\}}))$. The $f$ is related to the concrete Bayesian update rule for different distributions, and $f(\mathbf{y}_{\{1:t-1\}})$ is also notated as $\boldsymbol{\theta}_{t-1}$. Putting Eq. 2 into Eq. 1, the ELBO of BFNs could be further expressed as:

$$\mathcal{L}(\mathbf{x}) = -\mathcal{L}_{\text{ELBO}} = \sum_{t=1}^{n} D_{KL} \left( q(\mathbf{y}_t|\mathbf{x}) \| p_\phi(\mathbf{y}_t|\boldsymbol{\theta}_t = f(\mathbf{y}_{\{1:t-1\}})) \right) - \log p_O(\mathbf{x}|\boldsymbol{\theta}_n, \phi) \tag{3}$$

The term $\log p_O(\mathbf{x}|\boldsymbol{\theta}_n, \phi)$ is also known as reconstruction loss, which is directly computed over the network prediction with $\boldsymbol{\theta}_n$. As the term is generally negligible as $\boldsymbol{\theta}_n$ is very close to $x$, we do not involve the term in training loss following (Graves et al., 2023).

## 3 METHODOLOGY

In this section, we first identify the key difference between the GraphBFN and discrete diffusion-based approaches and discuss how the GraphBFN could generalize smooth dynamics of topological information for graph generation. Then, we provide a detailed formulation of our GraphBFN, along with a visualization of the training and testing procedure in Fig. 2. Lastly, we delve into several innovations for successfully realizing the concept and boosting the applications in graph generation.

## 3.1 Smoothing Topological Transformation via Infinite Noisy Discrete Graphs

We consider the following specific case of the perturbation distribution over discrete variables:

$$p_{\text{noise}}(\mathbf{y}_t \mid \mathbf{x}) = \text{Cat}(\mathbf{y}_t; \omega_t \mathbf{x} + (1 - \omega_t)\boldsymbol{\pi}) \quad (4)$$

Hence $\mathbf{x}$ refers to the one-hot representation over the $K$ categories and $\pi$ is the uniform prior over the categories, *i.e.* $[\frac{1}{K}, \cdots, \frac{1}{K}]$. For the discrete diffusion model (Vignac et al., 2022; Austin et al., 2021), the Eq. 4 could be seen as continuous-time variant of the variational distribution $q(\mathbf{y}_t|\mathbf{x})$ in Eq. 1, *i.e.* the extension of discrete-time uniform transition diffusion (Austin et al., 2021). In this scenario, the latent variable $y_t$ exists in a discrete space and is generated through random substitution or deletion of elements from the clean sample $x$, with a probability of $1 - \omega_t$. In the context of graphs, these operations can significantly alter the topology; for example, deleting a single edge in the adjacency matrix may impact the entire graph's connectivity.

Bayesian Flow Networks introduce a different concept of latent $\mathbf{y}_t$: Given the noisy distribution $p_{\text{noise}}$, we assume sample noisy samples from the noise distribution for $m$ times instead of once, thus incurring a count variable over the different categories as $\mathbf{c} = [c_1, \cdots, c_k]$. Based on the normalized counts $(\mathbf{c} - \frac{m}{K})$ and central limit theorem $\lim_{m \to \infty} = \frac{\mathbf{c} - mq_l}{\sqrt{mq_l}} \sim \mathcal{N}(0, \boldsymbol{I})$ (where $q_l = \omega_t \mathbf{x} + (1 - \omega_t)\boldsymbol{\pi}$), we could construct a continuous latent variable $\mathbf{y}$ with the k-th dimension as as:

$$y_k \stackrel{\text{def}}{=} \lim_{m \to \infty, \omega \to 0} \left(c_k - \frac{m}{K}\right) \ln(1 + \frac{\omega K}{1 - \omega}), \quad \text{we have } q(\mathbf{y}_t \mid \mathbf{x}; \alpha_t) = \mathcal{N}(\alpha_t(K\mathbf{x} - \mathbf{1}), \alpha K \boldsymbol{I}) \quad (5)$$

where $m\omega^2 = \alpha$ is a condition and $\alpha$ is a predefined finite number. The above $\mathbf{y}$ is obtained by assuming sampling **infinite samples** from the noise distribution. Detailed discussion is left as Appendix H.

In the context of graph generative modeling, the $\mathbf{x}$ could refer to the unit component of the graph, *e.g.* a node in the node vector ($\mathbf{V}_i$) or an edge entry in the adjacency matrix ($\mathbf{E}_{i,j}$). For topological analysis, we focus on the discussion of the adjacency matrix. We use $\mathbf{E}^{y_t}$ to denote the latent variable for the whole matrix $\mathbf{E}$, where the latent variable for each entry is therefore referred to as $\mathbf{E}_{i,j}^{y_t}$. Under the framework of GraphBFN, $\mathbf{E}_{i,j}^{y_t}$ is a count-based latent as shown in Eq. 5. Though the latent variable enjoys continuity and real-valued, it could be hard to view the $\mathbf{E}^{y_t}$ as valid adjacency matrices and analyze the topological dynamics from the perspective of the latent variable. Fortunately, we could instead focus on the parameter variable $\boldsymbol{\theta}_t$ which is the aggregation of the latent variable as introduced in Sec. 2.2. The aggregation $f$ in Eq. 3 for the discrete variable is defined by applying the following update rule recursively:

$$\theta_t = h(\theta_{t-1}, y_t) = \frac{e^{y_t}\theta_{t-1}}{\sum_{k=1}^{K} e^{y_t^{(k)}}\theta_{t-1}^{(k)}} \quad (6)$$

Here $\boldsymbol{\theta}$ lies in the probability simplex, i.e $\sum_{i=1}^{K}\boldsymbol{\theta}^k = 1$. In the above adjacency matrix setting, we use $\mathbf{E}_{i,j}^{\theta_t}$ to denote the corresponding parameter. Given $\mathbf{E}_{i,j}^{\theta_t}$ is on the probability simplex, it could be naturally interpreted as the probability over whether there is an edge between node $i$ and node $j$. We design a *Probabilistic Adjacency Matrix* $P$ based on $\mathbf{E}_{i,j}^{\theta_t}$: we force $\mathbf{E}^\theta$ to symmetric by $\mathbf{E}^\theta = (\mathbf{E}^\theta + (\mathbf{E}^\theta)^T)/2$; then, we derive the probability of whether there exist an edge at $E_{i,j}$ from $\mathbf{E}_{i,j}^\theta$ and make it as the corresponding value for the $P_{i,j}$. Under this definition of $P$ each entry $P_{i,j} \in [0, 1]$ could be seen as the parameter of the Bernoulli distribution.

*Remark* 3.1. The graph spectrum, which is defined as the eigenvalues or eigenvectors of the adjacency matrix or Laplacian matrix could be directly extended by viewing $P$ as the generalization of the adjacency matrix. We leave the detailed discussion over the graph spectrum in Appendix B.

The probabilistic adjacency matrix can be viewed as a special case of random graphs as in Athreya et al. (2018), providing bounds between the graph spectrum of sampled discrete graphs from $P$ and the generalized spectrum directly over $P$ (Chung & Radcliffe, 2011). The theoretical discussion is in Appendix C. The generation process in GraphBFN defines a transformation from $\mathbf{E}^{\theta_0}$ to $\mathbf{E}^{\theta_n}$, interpolating between $P^{\theta_0}$, a prior probabilistic matrix with all entries of $0.5$, and $P^{\theta_n}$, representing authentic graphs with binary entries. This interpolation enables GraphBFN to smoothly decompose complex discrete structures, as illustrated in Fig. 1. The detailed formulation of interpolation and training objectives will be introduced in the following section.

Figure 2: Diagram of GraphBFN. The solid lines and dashed lines denote the inference and training procedures, respectively. We note that $\Delta_\phi = \|\phi(\mathbf{G}^{\theta_t}, t) - \phi(\mathbf{G}^{\theta_{t-1}}, t-1)\|^2$.

### 3.2 GRAPH BAYESIAN FLOW NETWORKS

We elaborate on the components introduced in Sec. 2.2 to get a complete formulation of GraphBFN. Recall that the graph variable is denoted as $\mathbf{G} = (\mathbf{V}, \mathbf{E})$. Correspondingly, the latent variable at time step $t$ is referred to as $\mathbf{G}^{y_t} = (\mathbf{V}^{y_t}, \mathbf{E}^{y_t})$ and the corresponding aggregation parameter $\mathbf{G}^{\theta_t} = (\mathbf{V}^{\theta_t}, \mathbf{E}^{\theta_t})$. Then, we revisit the ELBO training objective introduced in Eq. 3. To improve conciseness, we use the adjacency matrix $\mathbf{E}$ as the primary example for derivation. Below, we illustrate the concrete formulation for the two key elements in the Eq. 3, $q\left(\mathbf{y}_t \mid \mathbf{x}\right)$ and $p_\phi\left(\mathbf{y}_t \mid \boldsymbol{\theta}_t = f\left(\mathbf{y}_{\{1:t-1\}}\right)\right)$.

**Sender distribution**  The $q\left(\mathbf{y}_t \mid \mathbf{x}\right)$ in the context of BFN is termed sender distribution. To define the sender distribution, we need extra involve an accuracy parameter $\alpha_t$ as also mentioned above in Eq. 5. Note the sender distribution for the adjacency matrix is set as independent for different entries, then the sender distribution here takes the form of

$$q\left(\mathbf{E}^{y_t} \mid \mathbf{E}, \alpha_t\right) = \Pi_{1 \le i,j \le n} q\left(\mathbf{E}_{i,j}^{y_t} \mid \mathbf{E}_{i,j}, \alpha_t\right) = \Pi_{1 \le i,j \le n} \mathcal{N}\left(\mathbf{E}_{i,j}^{y_t} \mid \alpha_t\left(c_e \mathbf{E}_{i,j} - \mathbf{1}\right), \alpha_t c_e \mathbf{I}\right) \quad (7)$$

**Receiver distribution**  $p_\phi\left(\mathbf{y}_t \mid \boldsymbol{\theta}_t = f\left(\mathbf{y}_{\{1:t-1\}}\right)\right)$ here is also named as the receiver distribution. Intuitively, $p_\phi\left(\mathbf{y}_t \mid \boldsymbol{\theta}_t\right)$ could be decomposed into first predict a distribution of $x$ with neural network $\phi$ based on $\theta_t$ and combine predicted distribution with the sender distribution above to get a distribution over $\mathbf{y}$. In our case, the receiver distribution takes the form of:

$$p_\phi(\mathbf{E}^{y_t} \mid \mathbf{G}^{\theta_t}) = \Pi_{1 \le i,j \le n} p_\phi(\mathbf{E}_{i,j}^{y_t} \mid \mathbf{G}^{\theta_t})$$

$$p_\phi(\mathbf{E}_{i,j}^{y_t} \mid \mathbf{G}^{\theta_t}) = \sum_{k=1}^{c_e} \mathbf{E}_{i,j}[\phi(\mathbf{G}^{\theta_t}, t)]^{(k)} \mathcal{N}\left(\mathbf{E}_{i,j}^{y_t} \mid \alpha_t\left(c_e \mathbf{e}_k - \mathbf{1}\right), \alpha_t c_e \mathbf{I}\right) \quad (8)$$

Here $\mathbf{e}_k$ is the one-hot vector with $k$-th dimension as 1 and the other $c_e - 1$ dimensions are all 0. Given the $\mathbf{G}^{\theta_t}$ and $t$ as network input, here $\mathbf{E}_{i,j}[\phi(\mathbf{G}^{\theta_t}, t)]^{(k)}$ is the probability on $k$-th category of the predicted categorical distributions for $\mathbf{E}_{i,j}$. Note the network $\phi$ usually has the final layer as softmax and hence output could be naturally seen as the probability of the predicted distribution, which is also referred to as output distribution.

**Simulation-free Training**  However, as mentioned above and also shown in Eq. 3, the network input is $\boldsymbol{\theta}_t$ which is determined by $\mathbf{y}_1, \cdots, \mathbf{y}_n$ and the discrete update rule in Eq. 6. Fortunately, with the discrete update rule, we could do aggregation in the distribution level for $q(\mathbf{y}_1|\mathbf{x}), \cdots, q(\mathbf{y}_n|\mathbf{x})$ and directly obtain the analytic form of distribution over $\boldsymbol{\theta}_t$, i.e. $q(\boldsymbol{\theta}_t|\mathbf{x})$. With the notation of $\mathbf{E}$, the distribution could be illustrated as:

$$q\left(\mathbf{E}_{i,j}^{\theta_t} \mid \mathbf{E}_{i,j}, \mathbf{E}_{i,j}^{\theta_0}\right) = \mathbb{E}_{N\left(\mathbf{E}_{i,j}^{y} \mid \beta(t)(c_e \mathbf{E}_{i,j} - \mathbf{1}), \beta(t) c_e \mathbf{I}\right)} \delta\left(\mathbf{E}_{i,j}^{\theta_t} - \frac{e^{\mathbf{E}_{i,j}^{y}} \odot \mathbf{E}_{i,j}^{\theta_0}}{\sum_{k=1}^{c_e}\left(e^{\mathbf{E}_{i,j}^{y}}\right)_k \left(\mathbf{E}_{i,j}^{\theta_0}\right)_k}\right) \quad (9)$$

where $\beta(t) = \sum_{j=0}^{t} \alpha_j$. Note that $\odot$ denotes the element-wise product between the vectors, and in $e^{\mathbf{E}_{i,j}^{y}}$, the exponential function is applied to each dimension individually. Such distribution is also noted as the Bayesian flow distribution. Putting Eq. 9, Eq. 8 and Eq. 7 in Eq. 3, we could obtain the final training objective.

**Continuous-time Training Objective**  The above discussion is based on that the number of latent variables is $n$, which is finite. It could also be possible to put $n \to \infty$, which could correspond to the continuous-time training. For convenience, we consider the value of timestep $t$ in $[0, 1]$. In this case, $\beta(t) = \int_0^t \alpha(t)$. For flexibility, we could use different scheduler of $\alpha(t)$ for edge and node, *i.e.* $\alpha_e(t)$ and $\alpha_v(t)$. With the following notations as:

$$q\left(\mathbf{E}^{\theta_t} \mid \mathbf{E}, \mathbf{E}^{\theta_0}\right) = \Pi_{1 \leq i,j \leq n} q\left(\mathbf{E}_{i,j}^{\theta_t} \mid \mathbf{E}_{i,j}, \mathbf{E}_{i,j}^{\theta_0}\right) \quad , \quad q\left(\mathbf{V}^{\theta_t} \mid \mathbf{V}, \mathbf{V}^{\theta_0}\right) = \Pi_{1 \leq i \leq n} q\left(\mathbf{V}_i^{\theta_t} \mid \mathbf{V}_i, \mathbf{V}_i^{\theta_0}\right)$$

$$q\left(\mathbf{G}^{\theta_t} \mid \mathbf{G}, \mathbf{G}^{\theta_0}\right) = q\left(\mathbf{E}^{\theta_t} \mid \mathbf{E}, \mathbf{E}^{\theta_0}\right) q\left(\mathbf{V}^{\theta_t} \mid \mathbf{V}, \mathbf{V}^{\theta_0}\right) \tag{10}$$

Under the continuous-time steps, there is a very concise analytic form for the KL divergence between the sender distribution (Eq. 7) and the receiver distribution (Eq. 8) over time step $t$ as:

$$L(\mathbf{E}, t) = c_e \frac{1}{2} \alpha_e(t) \underset{q\left(\mathbf{G}^{\theta_t} \mid \mathbf{G}, \mathbf{G}^{\theta_0}\right)}{\mathbb{E}} \sum_{1 \leq i,j \leq n} \left\| \mathbf{E}_{i,j} - \mathbf{E}_{i,j}[\phi(\mathbf{G}^{\theta_t}, t)] \right\|^2$$

$$L(\mathbf{V}, t) = c_v \frac{1}{2} \alpha_v(t) \underset{q\left(\mathbf{G}^{\theta_t} \mid \mathbf{G}, \mathbf{G}^{\theta_0}\right)}{\mathbb{E}} \sum_{1 \leq i \leq n} \left\| \mathbf{V}_i - \mathbf{V}_i[\phi(\mathbf{G}^{\theta_t}, t)] \right\|^2$$

Combining the above terms, our final objective for GraphBFN is as:

$$L(\mathbf{G}) = L\big((\mathbf{E}, \mathbf{V})\big) = \underset{t \sim U(0,1)}{\mathbb{E}} L(\mathbf{E}, t) + L(\mathbf{V}, t) \tag{11}$$

The detailed training and sampling algorithms are illustrated in Appendix E. And with perturbation equivariant architecture for $\phi$, *e.g.*, Graph Transformer, we have following properties:

**Proposition 3.2.** *If $\phi$ is a permutation equivariant function, then for any permutation $\pi$, we have:*

- *The objective in Eq. 11 is permutation invariant: $L(\mathbf{G}) = L((\mathbf{E}, \mathbf{V}) = L(\pi^T \mathbf{E} \pi, \pi^T \mathbf{V})$*

- *The density function, $p_\phi$, implied by the generative process of GraphBFN is also permutation invariant,* i.e., $p_\phi(\mathbf{E}, \mathbf{V}) = p_\phi(\pi^T \mathbf{E} \pi, \pi^T \mathbf{V})$

We leave the proof of the Proposition 3.2 in the Appendix I.

## 3.3 IMPROVED TECHNIQUES FOR GRAPHBFN

**Sampling with Adaptive Flowback**  The Bayesian Flow Networks conducted an autoregressive sampling procedure. For example, the $t$-th step vanilla sampling of BFN could be introduced as: first sample $\hat{G}_{t-1}$ according to the network output distribution $\phi(G^{\theta_{t-1}}, t-1)$; and assuming $\hat{G}$ as the final generated targets to sample $\hat{G}_{t-1}^{y_t}$ based on the sender distribution, and then apply the Bayesian update function to get $G^{\theta_t}$. However, there exists a notable discrepancy between training and sampling: During training the noisy samples $y$ are sampled based on real data; while at inference, the $y$ are sampled based on model prediction which brings extra errors, especially for the initial steps where there lacks information in the model inputs. The error will always be kept in the inputs due to the autoregressive Bayesian update, which could potentially hurt the performance.

To the end, we introduce another sampling procedure to make the sampling focus on the current prediction by using the Bayesian Flow distribution:

$$p_\phi\left(\mathbf{G}^{\theta_t} \mid \mathbf{G}^{\theta_{t-1}}\right) = q\left(\mathbf{G}^{\theta_t} \mid \phi(\mathbf{G}^{\theta_{t-1}}, t-1), \mathbf{G}^{\theta_0}\right) \tag{12}$$

$q\left(\mathbf{G}^{\theta_t} \mid \phi(\mathbf{G}^{\theta_{t-1}}, t-1), \mathbf{G}^{\theta_0}\right)$ here is the Bayesain flow distribution analogy to Eq. 9. We termed this approach as Flowback. Though the error accumulation is fixed, Flowback sampling could collapse due to that the network output distribution $\phi(\mathbf{G}^{\theta_t}, t)$ could hold low entropy practically. To balance exploration and exploitation, we combine the two sampling procedure and introduce the adaptive Flowback: with a predefined constant $\epsilon$, for any time $t$, when $\|\phi(\mathbf{G}^{\theta_t}, t) - \phi(\mathbf{G}^{\theta_{t-1}}, t-1)\|^2 \geq \epsilon$, we conduct vanilla sampling; when $\|\phi(\mathbf{G}^{\theta_t}, t) - \phi(\mathbf{G}^{\theta_{t-1}}, t-1)\|^2 < \epsilon$, the Flowback (Eq. 12) is conducted. Empirically, we find the adaptive Flowback could be approximately implemented by setting a threshold over $t$ to determine which kind of sampling to conduct to improve efficiency. The detailed sampling process is presented in Algorithm 2.

**Accuracy Scheduling for Any Prior** The accuracy scheduling is defined by the function $\beta(t) = \int_0^t \alpha(t)$, which represents the level the sender distribution corrupted the original information of the data samples. The original BFN paper (Graves et al., 2023) defines the accuracy scheduling as $\beta(t) = \beta(1)t^2$ and, correspondingly, the accuracy rate as $\alpha(t) = \beta(1)2t$. Such design relies on the intuition that, with a uniform prior, the expected entropy of $\boldsymbol{\theta}$ could linearly decrease along the Bayesian flow distributions. The scheduling also implies the fact that the accuracy is $0$ for the prior $\boldsymbol{\theta}_0$ as it is fully uninformative. However, in some scenarios of graph generation, an informative prior has been demonstrated to be effective (Vignac et al., 2022). To this end, we introduce a more general accuracy scheduling with a simple yet effective formulation:

$$\alpha(t) = a + 2\left(\beta(1) - a\right)t \quad \text{and} \quad \beta(t) = at + \left(\beta(1) - a\right)t^2 \tag{13}$$

Here $a$ is a hyperparameter. One degenerated case of Eq. 13 is when $a = \beta(1)$, which assigns a constant accuracy at each time. As shown in Fig. 3, Our general accuracy scheduling exhibits more steady paces of information reconstruction. This scheduling could be compatible with low-entropy prior, *i.e.* assigning a non-zero accuracy for informative prior, and also have a faster input entropy decrease.

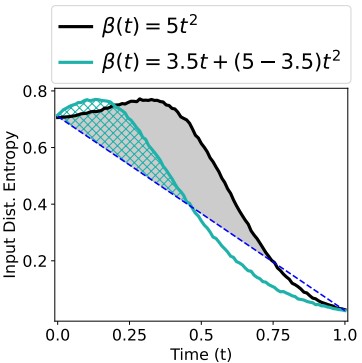

**Conditioned Graph Features on Output Distributions** Previous literature (Beaini et al., 2021; Vignac et al., 2022) has demonstrated that incorporating extra features of input graphs can enhance the representation power of graph neural networks. For example, Vignac et al. (2022) computes various graph descriptors of the noisy graph and includes them in the input of the denoising network during both training and sampling. In GraphBFN, we could also calculate the graph descriptors of the probabilistic adjacency matrix for the inclusion of the interdependency modeling network, *i.e.* $\phi$. We note that the extra

Figure 3: The entropy change of the input distribution $\mathbf{G}^{\theta_t}$ along the Bayesian flow equipped with a non-uniform prior.

graph feature is not necessarily used, and the choice is determined by the data property and complexity. For the usage of extra features, we apply a different strategy compared to previous literature (Vignac et al., 2022) by conditioning the network output distribution. We denote the module for computing the extra features as $M(\cdot)$, for example, obtaining the eigenvalues or eigenvectors of the adjacency matrix. The computation of the output distribution with the spectral feature conditioning is then formulated as:

$$\phi(\mathbf{G}^{\theta_t}, t) = \phi\left(\mathbf{G}^{\theta_t}, t, M(\text{sg}\,\phi(\mathbf{G}^{\theta_t}, t))\right) \tag{14}$$

where sg is the stop gradient operator. The key motivation of the proposed method is related to the self condition as proposed in Chen et al. (2022), which helps to improve the inner consistency of the generated samples. Such techniques are only used in large graphs such as the **SBM** dataset.

## 4 RELATED WORKS

From the fruitful lines of research conducted on generative modeling for graphs, we focus on the discussion over diffusion-based approaches as they are most related to our method. Recently, given the remarkable performance of diffusion models (Ho et al., 2020) in fields such as image generation, there has been a growing interest in graph diffusion models within the community. Research conducted by Jo et al. (2022); Niu et al. (2020); Luo et al. (2023) has successfully made the diffusion framework compatible with discrete graphs through the dequantization approach that adds continuous noise to discrete node and edge features. Jo et al. (2024) demonstrates the advantage of explicitly learning the final graph structures in the diffusion process; Vignac et al. (2022); Kong et al. (2023) highlights the benefits of modeling in the original discrete data space; Qin et al. (2023); Chen et al. (2023) leverage the spasity nature of graphs and show the possibility to scale the models up to larger graphs; Karami (2024) introduce hierarchical graph generative framework that captures the hierarchical nature of graphs. Furthermore, Vignac et al. (2022) highlights the potential issue of topology information getting distorted by continuous noise and thereby suggests utilizing more compatible discrete diffusions, resulting in noteworthy performance improvement.

In the literature of Bayesian Flow Network (Graves et al., 2023), there are several recent applications to the structured multi-modality data (Song et al., 2023; Qu et al., 2024) while the topological

Table 1: Main results on the general graph generation.

| | Planar | | | | | SBM | | | | |
| | Synthetic, $|V| = 64$ | | | | | Synthetic, $44 \leq |V| \leq 187$ | | | | |
| | Deg. ↓ | Clus. ↓ | Orbit ↓ | Spec. ↓ | V.U.N. ↑ | Deg. ↓ | Clus. ↓ | Orbit ↓ | Spec. ↓ | V.U.N. ↑ |
|---|---|---|---|---|---|---|---|---|---|---|
| Training set | 0.0002 | 0.0310 | 0.0005 | 0.0052 | 100.0 | 0.0008 | 0.0332 | 0.0255 | 0.0063 | 100.0 |
| GraphRNN | 0.0049 | 0.2779 | 1.2543 | 0.0459 | 0.0 | 0.0055 | 0.0584 | 0.0785 | 0.0065 | 5.0 |
| GRAN | 0.0007 | 0.0426 | 0.0009 | 0.0075 | 0.0 | 0.0113 | 0.0553 | 0.0540 | 0.0054 | 25.0 |
| SPECTRE | 0.0005 | 0.0785 | 0.0012 | 0.0112 | 25.0 | 0.0015 | 0.0521 | 0.0412 | 0.0056 | 52.5 |
| EDP-GNN | 0.0044 | 0.3187 | 1.4986 | 0.0813 | 0.0 | 0.0011 | 0.0552 | 0.0520 | 0.0070 | 35.0 |
| GDSS | 0.0041 | 0.2676 | 0.1720 | 0.0370 | 0.0 | 0.0212 | 0.0646 | 0.0894 | 0.0128 | 5.0 |
| DiGress | **0.0003** | 0.0372 | 0.0009 | 0.0106 | 75 | 0.0013 | 0.0498 | 0.0434 | 0.0400 | 74 |
| GruM | 0.0005 | 0.0353 | 0.0009 | 0.0062 | 90.0 | 0.0007 | **0.0492** | 0.0448 | **0.0050** | 85.0 |
| **GraphBFN** | 0.0005 | **0.0294** | **0.0002** | **0.0046** | **96.7** | **0.0005** | 0.056 | **0.037** | 0.0053 | **87.5** |

information modeling is less explored. Drawing inspiration from Graves et al. (2023), we propose the concept of probabilistic graph space, where the representation of topological information can be generalized, allowing for a smoother decomposition of information of discrete graphs. We leave a more detailed discussion over discrete diffusion models in the Appendix J.

## 5 EXPERIMENTS

In this section, we empirically study the effectiveness of the proposed framework. We conduct extensive experiments on both the abstract graph datasets and 2D molecule datasets to compare the performance of GraphBFN against several competitive graph generation baselines, including diffusion/score matching based methods, such as Digress (Vignac et al., 2022), GruM (Jo et al., 2024), GDSS (Jo et al., 2022) and EDP-GNN (Niu et al., 2020); latent variable models, such as JT-VAE (Jin et al., 2018). GAN-based method SPECTRE (Martinkus et al., 2022); Flow-based methods such as GraphAF (Shi et al., 2020), MoFlow (Zang & Wang, 2020) and GraphDF (Luo et al., 2021). Please refer to Appendix K for details about our experiment settings, the link to our implementation, and additional experiment results.

### 5.1 ABSTRACT GRAPH GENERATION

The abstract graph generation experiments test the model's ability to model complex graph topology. We conduct experiments on the benchmarks introduced in Vignac et al. (2022); Martinkus et al. (2022). It should be noted that we only need to model edge connections for the generation tasks of abstract graphs. The benchmark consists of two datasets: **Planar** with the constant number of 64 nodes per graph and **Stochastic Block Model (SBM)** with up to 200 nodes per graph. We evaluate the maximum mean discrepancy (MMD) of four graph statistics, *i.e.* degree (Deg.), clustering coefficient (Clus.), count of orbits with 4 nodes (Orbit), and the eigenvalues of the graph Laplacian (Spec.) between the generated graphs and test sets. Besides, we also report the percentage of valid, unique, and novel (V.U.N.) graphs, where a graph is considered valid if it satisfies the statistical properties of the **SBM** model or being planar and connected for the **Planar** dataset (Martinkus et al., 2022). The experiment result can be found in Tab. 1. GraphBFN has shown superior or competitive performance compared to all the baselines on both datasets. Specifically, on the Planar dataset, GraphBFN obtains a V.U.N. value approaching the limits, demonstrating the advantages of modeling the discrete graph in the probabilistic matrix space. We include more details of visualizations and analysis in Appendix Q and Appendix F.

### 5.2 2D MOLECULE GENERATIONS

For the molecule datasets, we conduct experiments on 2D molecule datasets with various scales: **QM9** which includes 100k molecules with up to 9 heavy atoms in the training set; **ZINC250k** which contains 250,000 drug-like molecules with a maximum atom number of 38; **MOSEs** with 1,936,962 drug-like molecules whose atom number is up to 38. GraphBFN jointly models the node features **V** and the graph typologies **E** of the molecular graphs. Besides, for the experiments on **QM9**, we consider both the without-hydrogen version and the more challenging version where the hydrogen will be explicitly modeled. For **QM9** with explicit hydrogen, we evaluate GraphBFN's performance

Table 2: Main results on the QM9 with *implicit* hydrogens and ZINC250k.

| Method | QM9 $(|V| \leq 9)$ | | | | ZINC250k $(|V| \leq 38)$ | | | |
|---|---|---|---|---|---|---|---|---|
| | Valid (%)↑ | FCD↓ | NSPDK↓ | Scaf.↑ | Valid (%)↑ | FCD↓ | NSPDK↓ | Scaf.↑ |
| Training set | 100.0 | 0.0398 | 0.0001 | 0.9719 | 100.0 | 0.0615 | 0.0001 | 0.8395 |
| MoFlow (Zang & Wang, 2020) | 91.36 | 4.467 | 0.0169 | 0.1447 | 63.11 | 20.931 | 0.0455 | 0.0133 |
| GraphAF (Shi et al., 2020) | 74.43 | 5.625 | 0.0207 | 0.3046 | 68.47 | 16.023 | 0.0442 | 0.0672 |
| GraphDF (Luo et al., 2021) | 93.88 | 10.928 | 0.0636 | 0.0978 | 90.61 | 33.546 | 0.1770 | 0.0000 |
| EDP-GNN (Niu et al., 2020) | 47.52 | 2.680 | 0.0046 | 0.3270 | 82.97 | 16.737 | 0.0485 | 0.0000 |
| GDSS (Jo et al., 2022) | 95.72 | 2.900 | 0.0033 | 0.6983 | 97.01 | 14.656 | 0.0195 | 0.0467 |
| DiGress (Vignac et al., 2022) | 98.19 | **0.095** | 0.0003 | 0.9353 | 94.99 | 3.482 | 0.0021 | 0.4163 |
| GruM | 99.69 | 0.108 | 0.0002 | **0.9449** | 98.65 | 2.257 | 0.0015 | 0.5299 |
| **GraphBFN** (Ours) | **99.73** | 0.101 | **0.0002** | 0.9386 | **99.22** | **2.116** | **0.0013** | **0.5304** |

Table 3: Main results on MOSES.

| Model | Class | Val ↑ | Unique↑ | Novel↑ | Filters↑ | FCD↓ | SNN↑ | Scaf↑ |
|---|---|---|---|---|---|---|---|---|
| VAE | SMILES | 97.7 | 99.8 | 69.5 | 99.7 | 0.57 | 0.58 | 5.9 |
| JT-VAE | Fragment | 100 | 100 | 99.9 | 97.8 | 1.00 | 0.53 | 10.0 |
| GraphINVENT | Autoreg. | 96.4 | 99.8 | – | 95.0 | 1.22 | 0.54 | 12.7 |
| ConGress | | 83.4 | 99.9 | 96.4 | 94.8 | 1.48 | 0.50 | 16.4 |
| DiGress | One-shot | 85.7 | 100 | 95.0 | 97.1 | 1.19 | 0.52 | 14.8 |
| **GraphBFN** | | 88.5 | 99.8 | 89.0 | 98.3 | 1.07 | **0.59** | 10.0 |

with two additional metrics, the Atom Stability, *i.e.* the percentage of the atoms with valid valency, and Molecule Stability for the percentage of generated molecules whose atoms all have valid valency. For other datasets, we follow the evaluation of previous works (Jo et al., 2024; Vignac et al., 2022) and report the corresponding metrics, including FCD, SNN, Scaffold similarity(Scaf), NSPDK, Novelty, and Validity, etc.

The experiment results of **QM9** can be found in Tab. 2 and Tab. 4. It could be found that GraphBFN shows superior or competitive performance compared with the previous method. In the challenging setting of **QM9** with explicit hydrogen, the GraphBFN could beat the baselines with a large margin and approach the empirical limits in the sense of molecule stability, which justifies the effectiveness of jointly modeling the distribution of node features and edge feature under the probabilistic parameter space.

The empirical results of **ZINC250k** and **MOSEs** in Tab. 2 and Tab. 3 show GraphBFN's consistently superior performance compared to the baselines, demonstrating its scalability to the large complex datasets. It should be noted that in Tab. 3, compared with other one-shot models, GraphBFN shows a slight decrease in novelty while maintaining a high uniqueness. One possible explanation lies in that the probability distribution learned with GraphBFN is so close to the training distribution that causes some overlap of the support. Despite this overfitting, the novelty still lies at a high level (89%). It also demonstrates the generalization ability of GraphBFN. We leave the visualization of generated samples in Appendix Q.

### 5.3 ABLATION STUDIES

**Sampling Steps** We conduct extensive ablation studies of GraphBFN. We first consider GraphBFN's property of sampling any steps and then examine the improved techniques we claimed in Sec. 3.3. One appealing property of GraphBFN is that with continuous-time objectives, the GraphBFN could sample with any steps. We conduct an ablation study of sampling steps on the dataset of **QM9** with *explicit* hydrogen and **Planar** in Tab. 4. It could be found that the GraphBFN with as few as 25 and 50 sampling steps could be on par with diffusion models with 500 and 1000 steps. This result implies a more than $10\times$ increase in sampling efficiency.

**Adaptive Flowback Sampling** We verified the effectiveness of the Adaptive Flowback sampling on the abstract graph dataset **Planar**. The experiment results can be found in Tab. 5. We note

Table 4: **Ablation study** on GraphBFN's ability to tradeoff sampling *quality* and *efficiency*. Our experiment on QM9 with *explicit* hydrogen and Planar demonstrates that GraphBFN can produce equally good samples as the previously best models with only 10% of their sampling steps (500 for QM9 and 1000 for Planar). A few entries are left as blank because the corresponding experiment results are presented or implemented in the original works (Vignac et al., 2022; Jo et al., 2024).

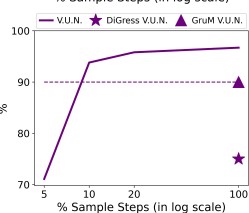

| | QM9 with H $|V| \leq 29$ | | | | Planar $|V| \leq 64$ |
|---|---|---|---|---|---|
| % Sample Steps | Valid (%) ↑ | Unique (%) ↑ | Atom Stab. (%) ↑ | Mol. Stab. (%) ↑ | V.U.N. (%) ↑ |
| 100 | **99.2** | 94.9 | **99.4** | **94.7** | **96.7** |
| 20 | 98.8 | 95.6 | 99.1 | 91.4 | 93.8 |
| 10 | 97.6 | 95.9 | 98.7 | 87.4 | 71.1 |
| 5 | 94.2 | 96.4 | 98.1 | 79.7 | 14.8 |
| 100 (DiGress) | 95.4 | **97.6** | 98.1 | 79.8 | 75.0 |
| 100 (GruM) | – | – | – | – | 90.0 |

Table 5: **Ablation study** on the impact of the adaptive Flowback threshold $\epsilon$ on generation *quality* and *diversity*. According to our experiment on Planar Graph, increaseing the use of Flowback sampling strategy always yields better sampling quality (as reflected in the strictly increasing V.U.N). However, overusing it might compromise sample diversity (as reflected in the sub-optimal graph statistics of $\epsilon = 1.0$ in comparison to $\epsilon = 0.5$).

| | Planar | | | | |
|---|---|---|---|---|---|
| $\epsilon$ | Deg. ↓ | Clus. ↓ | Orbit ↓ | Spec. ↓ | V.U.N. ↓ |
| Training Set | 0.0002 | 0.0310 | 0.0005 | 0.0052 | 100.0 |
| Flowback (1.0) | 0.0015 | 0.0703 | 0.0017 | 0.0047 | **98.2** |
| Adaptive Flowback (0.5) | **0.0005** | **0.0294** | **0.0002** | **0.0046** | 96.7 |
| Vanilla (0.0) | 0.0051 | 0.1793 | 0.0278 | 0.0099 | 14.1 |

that with vanilla sampling (i.e., $\epsilon = 0.0$, where the vanilla BFN sampling is used throughout the sampling process), the performance of V.U.N. is only $14.8\%$. With full Flowback, we could obtain a competitive performance with V.U.N. consistently outperforming the baseline, while the distribution-level metrics are still sub-optimal due to the collapse phenomenon. With adaptive Flowback, we could obtain the best quality-diversity tradeoff which makes adaptive Flowback a more appealing strategy.

Due to space limit, We defer the additional ablation studies on **Time Scheduler** and **Output Conditioned Feature** to Appendix G. Furthermore, we conducted an empirical study that quantifies the benefits of smooth graph topology transformation, with details presented in Appendix F.2.

## 6 CONCLUSIONS

We introduce GraphBFN for the generative modeling of discrete graphs. GraphBFN is motivated by the intuitive idea of smoothly decomposing and modeling the information of the graph topology by introducing a probabilistic matrix space. We first identify the train-test discrepancy issues in the vanilla formulation of Bayesian Flow Networks and propose several innovations under the framework. These appealing properties and improved techniques help GraphBFN achieve consistently superior performance on extensive graph generation benchmarks. Furthermore, GraphBFN can maintain decent sample quality while requiring much fewer sampling steps, thus largely improving the sample efficiency.

## ETHICS STATEMENT

We confirm that our work complies with the ICLR Code of Ethics, and we have carefully considered potential ethics concerns relating to the development and use of our proposed method GraphBFN for generative modeling of graph-structured data.

Our model is designed for general graph generation tasks and does not involve the use of sensitive personal data. However, we acknowledge that generative models, when applied to specific domains, such as molecular generation and social networking modeling, may inadvertently create synthetic data that could mimic private structures and details. Thus, we encourage users to apply our model in compliance with relevant privacy regulations and to critically evaluate its outputs.

We are committed to full transparency in our research. All datasets used in our experiments are publicly available and documented in detail to ensure reproducibility and compliance with legal standards. The methods and assumptions made in our research are clearly documented, and we provide substantial reproduction details in Appendix K. Furthermore, there is no conflict of interest, financial or otherwise, that could influence the development or presentation of this work.

Based on these considerations, we do not anticipate any violations of the ICLR Code of Ethics through the development or application of this model. However, we emphasize again that GraphBFN should not be used for malicious purposes, such as generating misleading data or creating harmful content.

## REPRODUCIBILITY STATEMENT

To ensure the reproducibility of our work, we provide a thorough description of the proposed Graph Bayesian Flow Networks (GraphBFN) in the main text, including details on the model architecture, training procedures, and evaluation metrics. All datasets, hyperparameters, and implementation details are clearly specified in both the main paper and Appendix K. Additionally, we will release our code along with instructions for reproducing the experiments upon the reception of the review result.

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

## A   DETAILED INTRODUCTION TO BAYESIAN FLOW NETWORKS

In this section, we introduce the basic components of Bayesian Flow Networks (BFNs) (Graves et al., 2023). The graphical model of BFN could be expressed in the form of a latent variable model whose prior is non-Markov autoregressive, distinguishing it from the diffusion models that have Markov priors. That is, for learning the probability distribution $p_\phi$ over sample space $\mathbf{x}$, a series of noisy versions $\mathbf{y}_{\{1:n\}} = \langle \mathbf{y}_1, \cdots, \mathbf{y}_n \rangle$ of $\mathbf{x}$ are introduced as latent variables. Then the model seeks to optimize the *variational lower bound* of the likelihood:

$$\log p_{\boldsymbol{\theta}}(\mathbf{x}) \geq \mathop{\mathbb{E}}_{\mathbf{y}_{\{1:n\}} \sim q} \left[ \log \frac{p_\phi(\mathbf{x}|\mathbf{y}_{\{1:n\}}) p_\phi(\mathbf{y}_{\{1:n\}})}{q(\mathbf{y}_{\{1:n\}}|\mathbf{x})} \right] = -D_{KL}(q \| p_\phi(\mathbf{y}_{\{1:n\}})) + \mathop{\mathbb{E}}_{\mathbf{y}_{\{1:n\}} \sim q} \log \left[ p_\phi(\mathbf{x} \mid \mathbf{y}_{\{1:n\}}) \right]$$

(15)

The variational distribution $q$ is defined by so-called sender distribution $p_S(\mathbf{y}_i \mid \mathbf{x}; \alpha_i)$. The sender distribution could be understood as adding noise to the data sample $\mathbf{x}$ with a predefined accuracy $\alpha_i$. The variational distribution is the fixed mean-field combination of sender distribution by $q(\mathbf{y}_{\{1:n\}} \mid \mathbf{x}) = \prod_{i=1}^n p_S(\mathbf{y}_i \mid \mathbf{x}; \alpha_i)$.

For the autoregressive prior $p_\phi(\mathbf{y}_1, \ldots, \mathbf{y}_n)$, the joint distribution could be extended with Bayesian decomposition as:

$$p_\phi(\mathbf{y}_1, \ldots, \mathbf{y}_n) = p_\phi(\mathbf{y}_1) \prod_{t=2}^n p_\phi(\mathbf{y}_i \mid \mathbf{y}_{\{1:t-1\}})$$

(16)

Note in BFN, the $\mathbf{y}$ is non-Markov contrast to diffusion models. In BFN, a new variable is introduced based on a fix-form functional transformation $f$, *e.g.* a weighted average function, as $\boldsymbol{\theta}_t = f(\mathbf{y}_{\{1:t\}})$ with $t > 1$ and $\boldsymbol{\theta}_0$ as a constant. We elaborate more on the definition of $t$, which is implied by successive Bayesian updates as $\boldsymbol{\theta}_t = h(\boldsymbol{\theta}_{t-1}, \mathbf{y}_t, \alpha_t)$. Given the constant prior $\boldsymbol{\theta}_0$ and a series noisy versions $\mathbf{y}_{\{1:n\}}$ of $\mathbf{x}$, we could obtain $\boldsymbol{\theta}_{\{1:n\}}$ through a sequence of Bayesian updates. The only randomness of the above process comes from $\mathbf{y}$, hence it could be summarized as a deterministic function $f$ over $\mathbf{y}$. $\boldsymbol{\theta}$ has the statistical interpretation as the parameter of the factorized distribution over the sample spaces by $p(\mathbf{x} \mid \boldsymbol{\theta}) = \prod_{d=1}^D p\left(\mathbf{x}^{(d)} \mid \boldsymbol{\theta}^{(d)}\right)$. The conditional distribution $p_\phi(\mathbf{y}_t \mid \mathbf{y}_{\{1:t-1\}})$ is defined as $p_\phi(\mathbf{y}_t \mid \mathbf{y}_{\{1:t-1\}}) = p_\phi(\mathbf{y}_t \mid \boldsymbol{\theta}_{t-1} = f(\mathbf{y}_{\{1:t-1\}}))$. Thus, Eq. 16 could be rewritten as: $p_\phi(\mathbf{y}_1, \ldots, \mathbf{y}_n) = \prod_{t=1}^n p_\phi(\mathbf{y}_t \mid \boldsymbol{\theta}_{t-1})$. We then clarify $p_\phi(\mathbf{y}_t \mid \boldsymbol{\theta}_{t-1})$. To obtain this term, we will first input $\boldsymbol{\theta}_{t-1}$ into the neural network $\Phi$ to capture the interdependency between different dimensions. Hence the distribution implied by $\boldsymbol{\theta}_{t-1}$, the neural network input, is also named as *input distribution*. The output of the neural networks denoted as $\hat{\boldsymbol{\theta}}_{t-1} = \Phi(\boldsymbol{\theta}_{t-1})$ still lies in the parameter space and the corresponding distribution is usually referred to as *output distribution*. Combining the formulation and accuracy $\alpha_t$ of *sender distribution* $p_S$, the distribution $p_\phi(\mathbf{y}_t \mid \boldsymbol{\theta}_{t-1})$ is defined as

$$p_\phi(\mathbf{y}_t \mid \boldsymbol{\theta}_{t-1}) = \mathbb{E}_{p_O(\hat{\mathbf{x}} \mid \Phi(\boldsymbol{\theta}_{t-1}))} p_S(\mathbf{y}_t \mid \hat{\mathbf{x}}; \alpha_t)$$

(17)

Note the $p_\phi(\mathbf{y}_t \mid \boldsymbol{\theta}_{t-1})$ is also formulated as *receiver distribution* $p_R(\mathbf{y}_t \mid \boldsymbol{\theta}_{t-1}; \Phi, \alpha_t)$.

Integrate Eq. 17 to Eq. 15, with $p_\phi(\mathbf{x}|\mathbf{y}_{\{1:n\}}) = p_O(\mathbf{x}|\boldsymbol{\theta}_n)$, we could get the objective as:

$$\mathcal{L}_{\text{ELBO}} = \sum_{t=1}^n D_{KL}(p_S(t) \| p_R(t)) + \log p_O(\mathbf{x}|\boldsymbol{\theta}_n)$$

(18)

Here $p_S(t)$ and $p_R(t)$ is abbreviation for $p_S(\mathbf{y}_t|\mathbf{x}, \alpha_t)$ and $p_R(\mathbf{y}_t \mid \boldsymbol{\theta}_{t-1}; \Phi, \alpha_t)$. To conduct simulation-free training on the ELBO, Graves et al. (2023) proposed to use the so-called Bayesian flow distribution $p_F(\boldsymbol{\theta}_{\{0:n\}}|\mathbf{x})$ formulated as $p_F(\boldsymbol{\theta}_t|\mathbf{x}) = \mathop{\mathbb{E}}_{\prod_{k=1}^t p_S(\mathbf{y}_k|\mathbf{x}; \alpha_k)} f(\mathbf{y}_{\{1:t\}})$, which allows for teacher-forcing-like training and generally has an analytical formulation.

## B   SPECTRAL GRAPH THEORY

Following previous literature (Vignac et al., 2022; Martinkus et al., 2022), we consider the definition of graph spectrum as the eigenvalues of the Laplacian matrix. For a graph $\mathbf{G}$ with $n$ nodes, the graph

Laplacian matrix is defined as $\mathbf{L} = \mathbf{D} - \mathbf{A}$, where $\mathbf{A}$ is the $n \times n$ adjacency matrix with $\mathbf{A}_{i,j} = 1$ if the node $v_i$ and node $v_j$ are connected and $\mathbf{A}_{i,j} = 0$ otherwise. $\mathbf{D}$ is the degree matrix, a diagonal matrix defined as $\mathbf{D} = \mathrm{diag}(d_1, \cdots, d_n)$. Here $d_i$ denotes the degree of the node $v_i$ as $d_i = \sum_{j=1}^{n} \mathbf{A}_{i,j}$. Besides, the normalized Laplacian Matrix $\mathbf{L}_{\mathrm{norm}}$ is defined as $\mathbf{L}_{\mathrm{norm}} = \mathbf{I} - \mathbf{D}^{-\frac{1}{2}} \mathbf{A} \mathbf{D}^{-\frac{1}{2}}$. For the symmetric positive semi-definite matrix $\mathbf{L}$ or $\mathbf{L}_{\mathrm{norm}}$, the eigen-decomposition is $\mathbf{U}\mathbf{\Lambda}\mathbf{U}^T$ where $\mathbf{U} = [\mathbf{u}_1, \cdots, \mathbf{u}_n]$ is an orthogonal matrix for the eigenvectors and $\mathbf{\Lambda} = \mathrm{diag}(\lambda_1, \cdots, \lambda_n)$ denotes the collection of eigenvalues. The eigenvalues and eigenvectors of the graph Laplacian imply some key properties of the graph topology structure, such as connectivity, clusterability, and distance between nodes (Grone et al., 1990).

**Spectral Gap and Graph Clusterability**   In this paragraph, we elaborate on the relationship between graph clusterability and the spectral gap that we've used to quantity graphs' topological information in Figure 1.

The spectral gap is computed by taking the difference between the first two smallest eigenvalues of graph matrices. Its definition varies slightly depending on whether the eigenvalues are taken from a graph's adjacency matrix or the Laplacian matrix. Nevertheless, regardless of which matrix it takes, the correlation holds that a smaller spectral gap implies poor graph connectivity and good clusterability. In our paper, we stick to the definition according to the *Laplacian Matrix*.

The Laplacian matrix $\mathbf{L}$ is symmetric and has non-negative and real eigenvalues. The multiplicity of 0 in the eigenvalues of $\mathbf{L}$ is equivalent to the number of connected components in the graph (Jiang, 2012).

Let $\lambda_1, \cdots, \lambda_n$ denote the eigenvalues of $L$ in the increasing order. The spectral gap in this scenario is defined as $\lambda_2 - \lambda_1$ (Martinkus et al., 2022). Since the smallest eigenvalue is always 0 (as the graph must have at least one connected component), the spectral gap is nothing but the second smallest eigenvalue of the Laplacian.

The second smallest eigenvalue is known as the Fiedler Value or the algebraic connectivity of a graph, and it reflects the connectivity of a graph  (Wyss-Gallifent, 2021). Specifically, given a fixed graph volume, the second smallest eigenvalue lowerbonds the diameter of a graph (Chung, 2006), which is a connectivity metric that measures the maximum distance between a pair of nodes in the graph. Therefore, if the second eigenvalue is small, the graph will contain nodes that are separated far apart from each other, implying loose connections and more clusters.

To gain more intuition for why a small spectral gap corresponds to a highly clustered graph, we can think about the extreme case where the spectral gap is 0. A zero spectral gap means that the second smallest eigenvalue of the Laplacian is also 0, implying the multiplicity of zero eigenvalues is at least 2. This means that the graph has at least 2 connected components (i.e. at least 2 clusters with no bridge between them).

## C   SPECTRAL FEATURES IN THE SPACE OF PROBABILISTIC ADJACENCY MATRICES

**Clarification on Probabilistic Adjacency Matrix Space**   Instead of focusing on the properties customized on the graph domain, we generalize the graph featured by its adjacency matrix to the more general matrix domains. The eigenvalues or eigenvectors of the matrices could be effective tools to reflect the essential properties. Then there are the following facts that are proved in (Chung & Radcliffe, 2011):

**Theorem C.1.** *(Properties and Bounds of PAM) For $D^p$ defined as the $\mathbb{E}_{A \sim A^p} D$, i.e. the expectation of the Degree matrix of the graph whose adjacency matrix $A$ sampled from $A^p$. Let $\delta$ be the minimum expected degree of $G$, and $L = I - D^{-1/2}AD^{-1/2}$ the (normalized) Laplacian matrix and similarly definition $L^p = I - (D^p)^{-1/2}A^p(D^p)^{-1/2}$. Choose $\epsilon > 0$:*

- *for $\Delta > \frac{4}{9}\ln(2n/\epsilon)$. Then with probability at least $1 - \epsilon$, for $n$ sufficiently large, the eigenvalues of $A$ and $A^p$ satisfy*

$$\forall 1 \leq j \leq n \, |\lambda_i(A) - \lambda_i(A^p)| \leq \sqrt{4\Delta \ln(2n/\epsilon)}$$

- *There exists a constant $k = k(\epsilon)$ such that if $\delta > k \ln n$, then with probability at least $1 - \epsilon$, the eigenvalues of $L$ and $L^p$ satisfy*

$$\forall 1 \leq j \leq n, |\lambda_j(L) - \lambda_j(L^p)| \leq 3\sqrt{\frac{3\ln(4n/\epsilon)}{\delta}}$$

The above theorem helps us connect the probabilistic matrix space to the graph space with theoretical justification. In particular, the Laplacian matrix could also get a similar definition in the probabilistic space whose eigenvalues could relate to the graph spectrum of sampled discrete graphs.

## D    BEYOND SIMPLE UNDIRECTED GRAPH

Our experiment shows that GraphBFN could handle the generation of undirected graphs. Furthermore, it could be naturally extended to other types of graphs, and we elaborate on how it could be adapted to them in the following.

**Directed graphs**    Compared to undirected graphs, directed graphs differ only in that they do not need the adjacency matrix to be symmetric. In our current implementation, we enforce the symmetricity on the network input $\mathbf{E}^{\theta_t}$ by the operation $\mathbf{E}^{\theta_t} = (\mathbf{E}^{\theta_t} + (\mathbf{E}^{\theta_t})^T)/2$, and on the final sample by $\mathbf{E}_{final} = (\mathbf{E}_{final} + \mathbf{E}_{final}^T)/2$. To study the directed graphs, we can simply remove these operations.

**Weighted graphs**    For weighted graphs where each edge in the graph is assigned a numerical value, the GraphBFN could still be applied by normalizing the numerical value to the [0,1] space. In this way, we could obtain a reformatted adjacency matrix, where an entry is set as the normalized numerical value if the corresponding edge exists and 0 otherwise. Then, we could apply the same objective of GraphBFN to such samples. It should be noted that the weighted graph is a hybrid of continuous and discrete variables, we could also use a hybrid GraphBFN for modeling, *i.e.* continuous BFN for modeling continuous variables and discrete for other discrete types like node and edge types.

**Attributed graphs**    When presenting our methodology, especially when we introduce the probabilistic adjacency matrix, we assume the edge type to be binary (existence and non-existence) for the simplicity of presentation. However, it could be noted that the GraphBFN framework could be easily extended to the attributed graphs where the node and edge have categorical attributes, e.g., more than two node/edge types. Furthermore, BFN's effectiveness on this general type of graph has been empirically verified in our experiments on molecule generation, where the graphical representations of molecules are attributed graphs. Lastly, we could still obtain the topological information under this setting by accumulating the probability of all categories other than the no-edge category to build the probabilistic adjacency matrix space.

## E    ALGORITHMS

---
**Algorithm 1** Discrete Variable Bayesian Flow
---

1: **Require:** $K \in \mathbb{R}$
    $K$ is the possible state number of a discrete variable
2: **function** BayesianFlow($\mathbf{x} \in \mathbb{R}^{D \times K}$, $\beta \in \mathbb{R}$)
3:     $\mathbf{y} \sim \mathcal{N}(\beta(K\mathbf{x} - \mathbf{1}), \beta K\mathbf{I})$
4:     $\boldsymbol{\theta} \leftarrow \text{softmax}(\mathbf{y})$
5:     **Return** $\boldsymbol{\theta}$
6: **end function**

---

---

**Algorithm 2** Sampling procedure with Adaptive Flowback

---

1: **Require:** $T, n \in \mathbb{Z}^+, c_v, c_e \in \mathbb{R}, \phi(\cdot), \beta_v(\cdot), \beta_e(\cdot), \alpha_v(\cdot), \alpha_e(\cdot), \epsilon$
    # $T$ denotes the sampling step, $N$ is the number of nodes, $c_v, c_e$ is the possible state of a node and edge variable, $\phi(\cdot)$ is the optimized neural network.$\beta$ and $\alpha$ are the corresponding scheduler functions.$\epsilon$ is the predefined constant

2: Initialize: $\mathbf{V}^\theta, \mathbf{E}^\theta$ with $\mathbf{V}^{\theta_0}, \mathbf{E}^{\theta_0}$

3: **for** $i = 1$ to $T$ **do**

4:      $t \leftarrow \frac{i-1}{T}$

5:      $\mathbf{V}[\phi(\mathbf{G}^{\theta_t}, t], \mathbf{E}[\phi(\mathbf{G}^{\theta_t}, t]] \leftarrow \phi(\mathbf{G}^{\theta_t} = (\mathbf{V}^{\theta_t}, \mathbf{E}^{\theta_t}), t)$

6:      $\mathbf{V}, \mathbf{E} \sim \text{Categorical}(\mathbf{V}[\phi(\mathbf{G}^{\theta_t}, t)), \text{Categorical}(\mathbf{E}[\phi(\mathbf{G}^{\theta_t}, t)])$

7:      $t_n = t + \frac{1}{T}, t_p = t - \frac{1}{T}$

8:      **if** $i > 2$ and $\|\phi(\mathbf{G}^{\theta_t}, t) - \phi(\mathbf{G}^{\theta_{t_p}}, t_p)\|^2 \geq \epsilon$ **then**

9:         $\mathbf{V}^{y_{t_n}}, \mathbf{E}^{y_{t_n}} \sim q\left(\mathbf{V}^{y_{t_n}} \mid \mathbf{V}, \alpha_v(t_n)\right), q\left(\mathbf{E}^{y_{t_n}} \mid \mathbf{E}, \alpha_e(t_n)\right)$

10:        $\mathbf{V}^{\theta_{t_n}}, \mathbf{E}^{\theta_{t_n}} \leftarrow \text{Bayesian Update}(\mathbf{V}^{y_{t_n}}, \mathbf{E}^{y_{t_n}}, \mathbf{V}^{\theta_t}, \mathbf{E}^{\theta_t})$ as Eq. 6

11:      **else**

12:         $\mathbf{V}^{\theta_{t_n}} \leftarrow \text{BayesianFlow}(\mathbf{V}[\phi(\mathbf{G}^{\theta_t}, t], \beta_v(t_n))$

13:         $\mathbf{E}^{\theta_{t_n}} \leftarrow \text{BayesianFlow}(\mathbf{E}[\phi(\mathbf{G}^{\theta_t}, t], \beta_e(t_n))$

14:      **end if**

15: **end for**

16: $\mathbf{V}[\phi(\mathbf{G}^{\theta_1}, 1)], \mathbf{E}[\phi(\mathbf{G}^{\theta_1}, 1)] \leftarrow \phi(\mathbf{G}^{\theta_t} = (\mathbf{V}^{\theta_1}, \mathbf{E}^{\theta_1}), 1)$

17: $\mathbf{V}, \mathbf{E} \sim \text{Categorical}(\mathbf{V}[\phi(\mathbf{G}^{\theta_1}, 1)), \text{Categorical}(\mathbf{E}[\phi(\mathbf{G}^{\theta_1}, 1)])$

18: **Return:** $\mathbf{V}, \mathbf{E}$

---

**Algorithm 3** Training procedure for one step

---

1: **Require:** function$\beta_v(t) = \int_0^t \alpha_v(t), \beta_e(t) = \int_0^t \alpha_e(t), c_v, c_e \in \mathbb{R}$, an NeuralNetwork $\phi(\cdot)$

2: **Input:** $\mathbf{V} \in \mathbb{R}^{n \times c_v}$ $\mathbf{E} \in \mathbb{R}^{n \times n \times c_e}, t \in [0, 1]$

3: $\mathbf{V}^{\theta_t} \leftarrow \text{BayesianFlow}(\mathbf{V}, t)$

4: $\mathbf{E}^{\theta_t} \leftarrow \text{BayesianFlow}(\mathbf{E}, t)$

5: $\mathbf{V}[\phi(\mathbf{G}^{\theta_t}, t)], \mathbf{E}[\phi(\mathbf{G}^{\theta_t}, t)] \leftarrow \phi(\mathbf{G}^{\theta_t} = (\mathbf{V}^{\theta_t}, \mathbf{E}^{\theta_t}), t)$

6: $L = c_e \frac{1}{2} \alpha_e(t) \sum_{1 \leq i,j \leq n} \left\| \mathbf{E}_{i,j} - \mathbf{E}_{i,j}[\phi(\mathbf{G}^{\theta_t}, t)] \right\|^2 + c_v \frac{1}{2} \alpha_v(t) \sum_{1 \leq i \leq n} \left\| \mathbf{V}_i - \mathbf{V}_i[\phi(\mathbf{G}^{\theta_t}, t)] \right\|^2$

7: Minimize loss $L$

---

# F ANALYSIS OF THE GENERATION PROCESS

## F.1 TRANSFORMATION OF GRAPH TOPOLOGY

GraphBFN and the diffusion-based models share the same advantage of gradually recovering the graph topological information in the generation process. However, the diffusion-based models experience more fluctuations in recovering graph topology. The instability arises because the denoising steps in these models are conditioned on the atomic samples $G^t$ represented in binary adjacency matrices, and such samples could involve significant noises and randomness. In contrast, in GraphBFN, the intermediate steps are conditioned on the input distribution $\mathbf{G}^{\theta_t}$ represented in the probabilistic adjacency matrix, which changes smoothly across the Bayesian update at each time step.

**Experiment Setup** To verify this discrepancy, we train GraphBFN and the diffusion-based models, Digress and Grums, to overfit a 2-cluster community graph with 20 nodes. The topological information of the graph is captured by the spectral gap, as it reflects clusterability—the most important topological feature of our overfitting sample. Recall that the spectral gap is calculated by taking the difference between the 1st and 2nd eigenvalues of the Laplacian for diffusion-based models and the probabilistic Laplacian for GraphBFN. The spectral gaps are measured on the intermediate noisy samples $E^t$ for DiGress and the input distribution $\mathbf{G}^{\theta_t}$ for GraphBFN, and, for GruM, we use the quantized $E_t$.

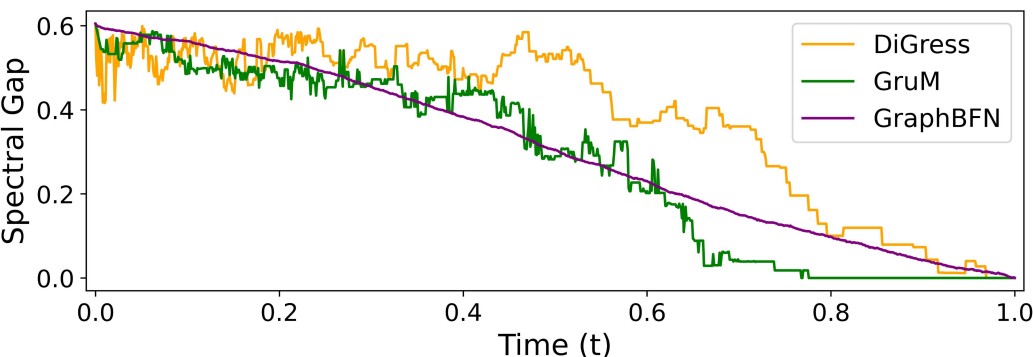

Figure 4: Topology recovery of GraphBFN.

Table 6: The smoothness of spectral gap transformation in the generation process of GraphBFN and the diffusion-based basline models

| Model | MASD(1e-2)↓ | MLSTD(1e-2)↓ | MVar(1e-2) ↓ |
|---|---|---|---|
| DiGress | 1.14 | 0.78 | 0.65 |
| GruM | 1.43 | 0.79 | 0.77 |
| **GraphBFN** | 0.39 | 0.25 | 0.26 |

**Results** We plot the changes of the spectral gaps in Figure. 4. In this figure, we see that GruM transforms information more smoothly than DiGress but not as smoothly as GraphBFN. Such observation is further confirmed by quantifying smoothness using the following metrics three metrics. The results of these metrics are shown in Table. 6.

- **Mean Absolute Second Derivative(MASD)**: MASD calculates the second derivative of the data, the 2nd derivative is then averaged out, and the equation is $S_{MASD} = \frac{\sum_{i=1}^{n-2}\left|\frac{d^2}{dx^2}(data_i)\right|}{n-2}$.

- **Mean Local Standard Deviation(MLSTD)**: MLSTD computes standard deviation in a small window, where, in our case, the window is set to 5 and then averages the local standard deviation. Intuitively, MLSTD captures relatively long-range smoothness in the window.

- **Mean Variation(MVar)**: MVar calculates the difference between neighboring data points, and we report the average. It emphasizes more on short-range smoothness.

F.2    ADVANTAGES OF "SMOOTH" TRANSFORMATION

We further conduct an empirical study that quantized the benefit of smooth graph topology transformation toward generation quality, thus revealing issues of diffusion-based graph generation models. We follow the same measure of smoothness described in the previous subsection.

**Experiment Setup** The toy overfitting experiments in the previous experiment does not reflect sample quality well, so in this experiment, we employ a more generalized setting of training on a complete dataset. In the first experiment, we train GraphBFN and DiGress (Vignac et al., 2022) on the complete **SBM** dataset (using the best configuration for both) and let each sample 200 graphs

Table 7: The study of the advantages of smoothness topology transformation at generation by comparing GraphBFN and the diffusion-based model DiGress. The smoothness metrics are in the scale of 1e-2

| Model | MLSTD↓ | MVar ↓ | Deg. ↓ | Clus. ↓ | Orbit ↓ | Spec. ↓ | V.U.N. ↑ |
|---|---|---|---|---|---|---|---|
| DiGress | 0.79 | 0.62 | 0.0015 | 0.0512 | 0.0396 | 0.0379 | 73.5 |
| **GraphBFN** | 0.23 | 0.25 | 0.0004 | 0.0509 | 0.0328 | 0.0050 | 90.0 |

Table 8: The study of the advantages of smoothness topology transformation at generation through the failure cases of the diffusion-based model DiGress. The smoothness metrics are in the scale of 1e-2.

| Model | MLSTD↓ | MVar ↓ | Deg. ↓ | Clus. ↓ | Orbit↓ | Spec. ↓ | V.U.N. ↑ |
|---|---|---|---|---|---|---|---|
| DiGress (original) | 0.79 | 0.62 | 0.0015 | 0.0512 | 0.0396 | 0.0379 | 73.5 |
| DiGress (highest 100) | 1.04 | 0.76 | 0.0018 | 0.0518 | 0.0461 | 0.0450 | 63 |
| DiGress (lowest 100) | 0.54 | 0.48 | 0.0007 | 0.0498 | 0.0336 | 0.0212 | 84 |

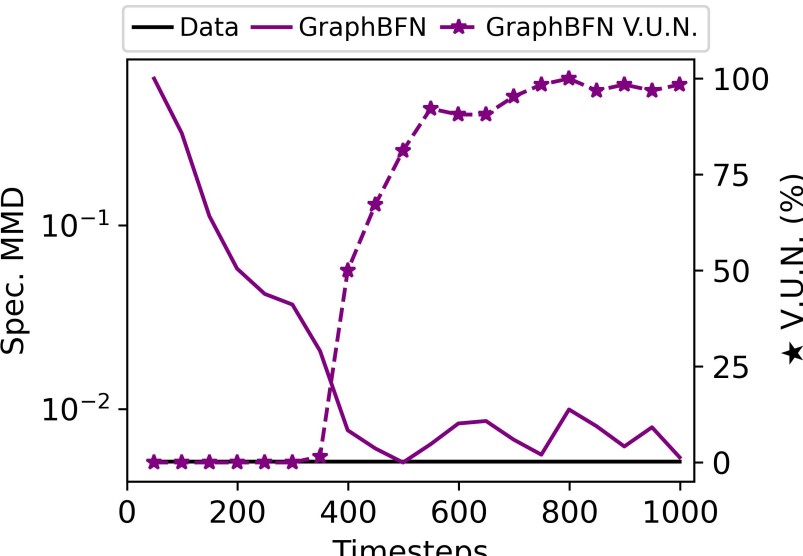

Figure 5: A comparison of the transformation of graph topology between GraphBFN and the diffusion-based models, DiGress and GruM.

with 1,000 sample steps. We then collect intermediate noise samples and compute the smoothness metrics MLSTD and MVar on the spectral gap as in Sec. F.1. For the final generated samples, we compute the sample quality metrics as in Tab. 1.

Furthermore, we investigate how smoothness in transformation correlates with failure cases diffusion-based approach. We equally partition the 200 samples generated by DiGress into two categories based on the smoothness metric MLSTD, with one group containing the samples with the bottom 100 MLSTD and the other containing those with the top 100.

**Results** According to the results of the above experiments, as shown in Tab. 7 and Tab. 8, it is evident that there is a significant performance gap between the sample groups with different smoothness metrics. By selecting the smoother transformation trajectories during generation, the performance of diffusion-based approaches can be significantly improved. This finding better justifies the issues of unstable dynamics in traditional diffusion-based methods.

## F.3 TOPOLOGY RECOVERY

We conducted a similar analysis as in Jo et al. (2024) on the recovery of graph topology in the generation process. The results of GraphBFN are shown in Figure. 5. Comparing to Figure.2 of Jo et al. (2024), we could find that GraphBFN shares with GruM the advantage of more speedy topology recovery, with GraphBFN recovering the graph topology even faster.

Table 9: **Ablation study** on the technique of conditioning GraphBFN's the training and generation processes on the extra graph features drawing from the PMA of GraphBFN's *output distribution*. Experiments on SBM and Planar graph datasets show that this technique boosts performance in terms of both generation quality and diversity.

| | Planar | | | | | SBM | | | | |
| | Synthetic, $|V| = 64$ | | | | | Synthetic, $44 \leq |V| \leq 187$ | | | | |
| | Deg. ↓ | Clus. ↓ | Orbit ↓ | Spec. ↓ | V.U.N. ↑ | Deg. ↓ | Clus. ↓ | Orbit ↓ | Spec. ↓ | V.U.N. ↑ |
|---|---|---|---|---|---|---|---|---|---|---|
| Training set | 0.0002 | 0.0310 | 0.0005 | 0.0052 | 100.0 | 0.0008 | 0.0332 | 0.0255 | 0.0063 | 100.0 |
| None | 0.0647 | 0.3525 | 1.695 | 0.0794 | 0.0 | 0.0149 | 0.0748 | 0.125 | 0.0116 | 0.0 |
| Input Dist. | 0.0006 | 0.0288 | 0.0013 | 0.0058 | 87.5 | 0.0009 | 0.058 | 0.054 | 0.0057 | 75.0 |
| Output Dist. | **0.0005** | **0.0294** | **0.0002** | **0.0046** | **96.7** | **0.0005** | **0.056** | **0.037** | **0.0053** | **87.5** |

Table 10: **Ablation study** on different accuracy scheduling of GraphBFN. According to our experiment on SBM, the scheduling with the general accuracy rate function $\alpha(t) = a + 2(\beta(1) - a)t$ outperforms that with the quadratic rate in the original paper of Graves et al. (2023).

| | SBM | | | | |
| | Synthetic, $|V| = 187$ | | | | |
| Acc. Sched. | Deg. ↓ | Clus. ↓ | Orbit ↓ | Spec. ↓ | V.U.N. ↓ |
|---|---|---|---|---|---|
| Training Set | 0.0008 | 0.0332 | 0.0255 | 0.0063 | 100.0 |
| $\alpha(t) = 2\beta(1)t$ | 0.046 | 0.0574 | 0.069 | 0.0081 | 75.0 |
| $\alpha(t) = a + 2(\beta(1) - a)t$ | **0.0005** | **0.056** | **0.037** | **0.0053** | **87.5** |

## G  ADDITIONAL ABLATION STUDIES

**Output Distribution Conditioning** We evaluate the role that the spectral features of graphs play in the generation process. We conduct an ablation study on SBM by training GraphBFN with and without conditioning on the features. When conditioning on the features, we consider extracting the spectral features from two sources, either from the PAM of the GraphBFN's input distribution or the output distribution. Results in Tab. 9 show that incorporating information on the spectral features is essential for correctly modeling the internal rules of larger graphs like SBM. Besides, the output distribution could be a better source to extract the spectral features as shown by the further boost in performance. This phenomenon is within our expectation because the input distribution models the distribution of each edge in a graph independently and thus offers less clear spectral information.

**Accuracy Scheduling** We experimented with how the accuracy scheduling of GraphBFN impacted its generation results. We set up the experiment on SBM, the abstract graph at scale, and trained GraphBFN with both the original quadratic scheduling and our generalized scheduling, with $\beta(1)$ set to 3.0 and $\alpha$ set to 1.0 for the generalized. Results in Tab. 10 show that the generalized scheduling could improve the model's performance.

## H  DERIVATION OF BFN OVER DISCRETE VARIABLE

To obtain the exact form of the sender distribution, the key is to apply the central limit theorem and take the condition of $m\omega^2 = \alpha$ and is finite, which implies $m \to \infty$ and $\omega \to 0$. Also with the first-order Taylor extension could obtain the final result. For a detailed derivation of the BFN over discrete variables, we would like to suggest referring to the Eq.148 to Eq.157 in (Graves et al., 2023).

## I  PROOF OF PROPOSITION 3.2

We prove the proposition in the following:

*Proof.* We first prove the loss in Eq. 11 is permutation invariant. To this end, we first prove that

$$q\left(\mathbf{G}^{\theta_t} \mid \mathbf{G}, \mathbf{G}^{\theta_0}\right) = q\left(\pi\mathbf{G}^{\theta_t} \mid \pi\mathbf{G}, \pi\mathbf{G}^{\theta_0}\right) \tag{19}$$

We take the same decomposition as in Eq. 10 of the term $q\left(\mathbf{G}^{\theta_t} \mid \mathbf{G}, \mathbf{G}^{\theta_0}\right)$:

$$q\left(\mathbf{E}^{\theta_t} \mid \mathbf{E}, \mathbf{E}^{\theta_0}\right) = \Pi_{1 \leq i,j \leq n} q\left(\mathbf{E}_{i,j}^{\theta_t} \mid \mathbf{E}_{i,j}, \mathbf{E}_{i,j}^{\theta_0}\right) \quad , \quad q\left(\mathbf{V}^{\theta_t} \mid \mathbf{V}, \mathbf{V}^{\theta_0}\right) = \Pi_{1 \leq i \leq n} q\left(\mathbf{V}_i^{\theta_t} \mid \mathbf{V}_i, \mathbf{V}_i^{\theta_0}\right)$$

$$q\left(\mathbf{G}^{\theta_t} \mid \mathbf{G}, \mathbf{G}^{\theta_0}\right) = q\left(\mathbf{E}^{\theta_t} \mid \mathbf{E}, \mathbf{E}^{\theta_0}\right) q\left(\mathbf{V}^{\theta_t} \mid \mathbf{V}, \mathbf{V}^{\theta_0}\right) \tag{20}$$

We take the $q\left(\mathbf{V}^{\theta_t} \mid \mathbf{V}, \mathbf{V}^{\theta_0}\right)$ as example:

$$q\left(\pi\mathbf{V}^{\theta_t} \mid \pi\mathbf{V}, \pi\mathbf{V}^{\theta_0}\right) = \pi(\Pi_{1 \leq i \leq n} q\left(\mathbf{V}_i^{\theta_t} \mid \mathbf{V}_i, \mathbf{V}_i^{\theta_0}\right)) = \Pi_{1 \leq i \leq n} q\left(\mathbf{V}_i^{\theta_t} \mid \mathbf{V}_i, \mathbf{V}_i^{\theta_0}\right)$$

$$= q\left(\mathbf{V}^{\theta_t} \mid \mathbf{V}, \mathbf{V}^{\theta_0}\right) \tag{21}$$

Similar derivation could be applied to $\mathbf{E}$. And hence we could finish the proof.

Next, we take the $L(\mathbf{V}, t)$ as an example to prove $L(\mathbf{V}, t) = L(\pi\mathbf{V}, t)$:

$$L(\pi\mathbf{V}, t) = c_v \frac{1}{2} \alpha_v(t) \mathbb{E}_{q\left(\pi\mathbf{G}^{\theta_t} \mid \pi\mathbf{G}, \pi\mathbf{G}^{\theta_0}\right)} \sum_{1 \leq i \leq n} \left\| \pi\mathbf{V}_i - \mathbf{V}_i \left[\phi\left(\pi\mathbf{G}^{\theta_t}, t\right)\right] \right\|^2 \tag{22}$$

For simplicity, we focus on the term related to $\pi$, as $\phi$ is permutation equivariant, we have

$$\mathbb{E}_{q\left(\pi\mathbf{G}^{\theta_t} \mid \pi\mathbf{G}, \pi\mathbf{G}^{\theta_0}\right)} \sum_{1 \leq i \leq n} \left\| \pi\mathbf{V}_i - \mathbf{V}_i \left[\phi\left(\pi\mathbf{G}^{\theta_t}, t\right)\right] \right\|^2 = \mathbb{E}_{q\left(\mathbf{G}^{\theta_t} \mid \mathbf{G}, \mathbf{G}^{\theta_0}\right)} \sum_{1 \leq i \leq n} \left\| \pi\mathbf{V}_i - \pi\mathbf{V}_i \left[\phi\left(\mathbf{G}^{\theta_t}, t\right)\right] \right\|^2$$

$$= \mathbb{E}_{q\left(\mathbf{G}^{\theta_t} \mid \mathbf{G}, \mathbf{G}^{\theta_0}\right)} \pi(\sum_{1 \leq i \leq n} \left\| \mathbf{V}_i - \mathbf{V}_i \left[\phi\left(\mathbf{G}^{\theta_t}, t\right)\right] \right\|^2) = \mathbb{E}_{q\left(\mathbf{G}^{\theta_t} \mid \mathbf{G}, \mathbf{G}^{\theta_0}\right)} \sum_{1 \leq i \leq n} \left\| \mathbf{V}_i - \mathbf{V}_i \left[\phi\left(\mathbf{G}^{\theta_t}, t\right)\right] \right\|^2$$

$$\tag{23}$$

Then we could also apply the same process to $\mathbf{E}$. Then we finish the proof over permutation invariance of the loss function.

Next, we discuss the invariant density modeling of the $p_\phi$. This could be directly following the conclusion of Theorem 3.1 of Song et al. (2023). With a permutation invariant prior $\mathbf{G}^{\theta_0}$ and a permutation equivariant interdependency network $\phi$, we only need to make sure the Bayesian Update function $h$ is permutation equivariant. As we could find in Eq. 6,

$$h(\pi\theta_{t-1}, \pi y_t) = \pi(\frac{e^{y_t}\theta_{t-1}}{\sum_{k=1}^K e^{y_t^{(k)}\theta_{t-1}^{(k)}}}) = \pi h(\theta_{t-1}, y_t) \tag{24}$$

And then we could finish the proof.

$\square$

## J  DISCUSSION WITH DISCRETE DIFFUSION MODELS

The key intuition held by Bayesian Flow Networks(BFN) (Graves et al., 2023) is that the information in the generative process should be modeled in a smoothly increasing fashion for superior performance. Such intuition is shared by the BFN and diffusion in the continuous data and is also identified by several studies under the literature of diffusion models for image generations (Ho et al., 2020). GraphBFN is motivated by the compatibility of such "continuous information modeling" intuition and the special nature of graph data: that is, though generally represented in the discrete modality, the essential graph topology information could also be represented in the continuous manifold as implied by graph spectrum. This allows GraphBFN to better harness the inductive bias in modeling discrete data modality with continuous natures, compared to the previous discrete diffusion models (Vignac et al., 2022). Furthermore, compared to the methods that directly dequantized the data and applied continuous diffusion models, the GraphBFN needs no quantize/dequantize procedure and could be more compatible with the sparse adjacency matrix representation.

There are also some potential empirical advantages of GraphBFN. The input to the neural network of GraphBFN lies in the continuous manifold, which could enjoy the advantages of numerical stability. More importantly, the inputs to the continuous distribution hold a considerably small variance compared to the previous discrete diffusion models (Graves et al., 2023). Also, such properties could lead to a speed-up in convergence during training.

## K    EXPERIMENT DETAILS AND ADDITIONAL RESULTS

### K.1    EXPERIMENT DETAILS

The datasets, dataset splits, and evaluation metrics of QM9 with explicit hydrogen and MOSES follow from Vignac et al. (2022), those of SBM, Planar, QM9 with implicit hydrogen, and ZINC250k follow from Martinkus et al. (2022), and those of Community-small, Ego-small, and Protein follow from Jo et al. (2022).

The network architectures and settings directly followed Vignac et al. (2022); Jo et al. (2024). The models are trained with the AdamW optimizer (Loshchilov & Hutter, 2019) until convergence. The most sensitive hyperparameters of GraphBFN are $\beta_{\mathbf{E}}$ and $\beta_{\mathbf{V}}$ that control the speed of information transmission in the generation process. We search them in the discrete linear space $\{1.0, 2.0, \cdots, 10.0\}$.

For all of our experiments, we use NVIDIA GeForce RTX 3090 with 24GB memory to train and evaluate our models. The batch sizes are adjusted to the maximum that can fit within the memory constraint.

All of our reported results are the average of three random runs. For a cleaner presentation and to follow the convention of previous work, we only add the error terms to the result of **QM9 with Explicit Hydrogen**.

For the results of **QM9 with Explicit Hydrogen** in Tab. 4, we only compare to one other method because, to our best knowledge, DiGress (Vignac et al., 2022) is the only previous graph generative model that has attempted to model QM9 molecules with *explicit* hydrogen.

**Extra Features**: We use extra features for the inputs of the GraphTransformer which directly follows (Vignac et al., 2022). Specifically, we have included cycle counts using pre-determined formulas to efficiently calculate cycles up to size 6. Node features count participation in up to 5-cycles, while graph features tally up to 6-cycles, utilizing node degree vectors and the Frobenius norm. Additionally, we integrate spectral features, which involve an $O\left(n^3\right)$ eigendecomposition—manageable for our graphs with up to 200 nodes. Key graph-level features derived from the graph Laplacian include the number of connected components and the first five nonzero eigenvalues. Node-level features estimate the largest connected component and include the first two eigenvectors associated with nonzero eigenvalues.

Each spectral feature or structural feature could be categorized into node-level features and graph-level features. Specifically, a node-level feature holds the shape of (batch-of-graphs, node-in-graph), such as an estimation of the biggest connected component (using the eigenvectors associated with eigenvalue 0) ; a graph-level feature holds the shape of (batch-of-graphs, 1). After computing the node-level features $F\_n$ and graph-level features $F\_g$ based on $V^\theta$ and $E^\theta$, we concat $V^\theta$ and all $F\_n$ as new node representations $V\_$feat. and concat all $F\_g$ as the finally graph-level feature $y$. Then the inputs for the graph transformers in GraphBFN are the triple of $(V\_$feat.$, E^\theta, y)$ similarly to Digress (Vignac et al., 2022)

**Scheduling Function:** In short, our scheduling design in Eq. 13 is an improvement upon the original design in vanilla BFN (Graves et al., 2023) that is determined empirically based on the intuitive of linearly increased entropy. Our design attempts to change the non-informative prior to the original design to be informative.

In Eq. 13, $\beta(t)$ is the accumulative(integral of $\alpha(t)$ from $0$ to $t$). The key intuition behind the design is that for informative prior, the initial step 0 already contains information of the ground truth sample. Then we set $\alpha(0)$ to a positive constant $a$. Besides, we also follow the heuristic of vanilla BFN (Graves et al., 2023) to use the square scheduler for accumulated accuracy $\beta$ to ensure the entropy of the input distribution approximately changes linearly along the timesteps. Eq. 13 is a very simple form that satisfies both properties.

## K.2 COMPUTATION EFFICIENCY

Training time for QM9 takes approximately 3 hours, and for large molecules and abstract datasets, the longest training takes 4 days.

For the inference efficiency, we conducted an experiment that compares GraphBFN with two other multi-step graph generation models – DiGress (Vignac et al., 2022) and GruM (Jo et al., 2024). We record the time for each model to sample 10000 graphs of QM9 with explicit hydrogen with the same sampling steps of 500. The results are shown in the Tab. 11.

Table 11: Comparing GraphBFN with other multi-step generation methods on the time to Sample 10,000 QM9 with *Explicit* Hydrogen Graphs.

| | Sampling Time (min.) |
| --- | --- |
| DiGress | 40 |
| GruM | 35 |
| **GraphBFN** | 44 |

According to the results, GraphBFN has approximately the same inference speed as the other two diffusion-based models. This result is expected, as all three methods require only one network evaluation at each iteration. GruM enjoys the fastest sampling speed. We speculate that this is because GruM operates in the space of continuous data, so it does not require the relatively costly procedure of sampling from categorical distributions.

Table 12: Additional results on smaller datasets. Most baseline performances are adapted from Jo et al. (2022) and Kong et al. (2023). Hyphens(-) denote unobtainable results due to either the reasons mentioned in Jo et al. (2022) or the lack of official implementation.

| | | Ego-small | | | Community-small | | | Enzymes | | |
| --- | --- | --- | --- | --- | --- | --- | --- | --- | --- | --- |
| | | Real, $4 \le |V| \le 18$ | | | Synthetic, $12 \le |V| \le 20$ | | | Real, $10 \le |V| \le 125$ | | |
| | | Deg.↓ | Clus.↓ | Orbit↓ | Deg.↓ | Clus.↓ | Orbit↓ | Deg.↓ | Clus.↓ | Orbit↓ |
| | Training set | 0.002 | 0.021 | 0.005 | 0.02 | 0.07 | 0.01 | 0.0003 | 0.01079 | 0.0003 |
| Autoreg. | DeepGMG | 0.040 | 0.100 | 0.020 | 0.220 | 0.950 | 0.400 | - | - | - |
| | GraphRNN | 0.090 | 0.220 | 0.003 | 0.080 | 0.120 | 0.040 | 0.017 | 0.062 | 0.046 |
| | GraphAF | 0.03 | 0.11 | 0.001 | 0.18 | 0.20 | 0.02 | 1.669 | 1.283 | 0.266 |
| | GraphDF | 0.04 | 0.13 | 0.01 | 0.06 | 0.12 | 0.03 | 1.503 | 1.061 | 0.202 |
| One-shot | GraphVAE | 0.130 | 0.170 | 0.050 | 0.350 | 0.980 | 0.540 | 1.369 | 0.629 | 0.191 |
| | GNF | 0.030 | 0.100 | 0.001 | 0.200 | 0.200 | 0.110 | - | - | - |
| | EDP-GNN | 0.052 | 0.093 | 0.007 | 0.053 | 0.144 | 0.026 | 0.023 | 0.268 | 0.082 |
| | GDSS | 0.021 | 0.024 | 0.007 | 0.045 | 0.086 | 0.007 | 0.026 | 0.061 | 0.009 |
| | SPECTRE | 0.078 | 0.078 | 0.007 | 0.020 | 0.210 | 0.010 | 0.136 | 0.195 | 0.125 |
| | DiGress | 0.015 | 0.029 | **0.005** | 0.020 | 0.063 | 0.010 | **0.004** | 0.083 | 0.002 |
| | EDGE | - | - | - | 0.008 | 0.011 | 0.026 | - | - | - |
| | GraphARM | 0.019 | **0.017** | 0.010 | 0.034 | 0.082 | **0.004** | 0.029 | 0.054 | 0.015 |
| | HiGen | - | - | - | - | - | - | 0.012 | 0.038 | **7.2e-4** |
| | **GraphBFN** | **0.0034** | 0.0246 | **0.0057** | **0.0075** | **0.0095** | **0.0043** | 0.0064 | **0.0212** | 0.0051 |

## K.3 ADDITIONAL RESULTS ON NON-ATTRIBUTED GRAPHS

In Tab. 12, we provide results on non-attributed (i.e. graph with only one edge type and node type) graph datasets, along with comparisons to additional baselines (Chen et al., 2023; Kong et al., 2023; Karami, 2024). **Community-small** contains 100 randomly generated clustered graphs; **Ego-small** contains 200 small ego graphs; and **Enzymes** contains 587 chain-like graphs representing protein structures (Jo et al., 2022).

According to the table, GraphBFN has shown superior performance on Community-small and competitive performance on Ego-small and Enzymes. It should be noted that for **Community-small** and **Ego-small**, the metrics are very saturated due to the limited complexity of datasets; as we can see, the SOTA results are very close to the training set's performance.

## K.4 CODE

More specific details about the experiment setting and our implementation of GraphBFN can be found in our code at https://anonymous.4open.science/r/GraphBFN-5FEE/README.md.

## L FLOWBACK SAMPLING

The visualization of GraphBFN's generation process with the Flowback technique is in Figure. 6.

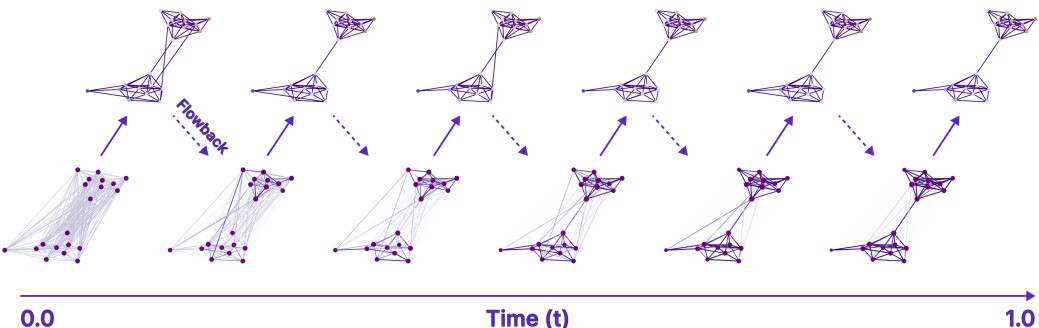

Figure 6: Visualization of GraphBFN's generation process with the Flowback technique. The sample graphs in the first row are sampled from the output distribution $\phi(\mathbf{G}^{\theta_t}, t)$ at different time steps, while those on the second line are representations of the probabilistic adjacency matrix of the input distribution $\mathbf{G}^{\theta_t}$. The solid lines denote the network prediction, while the solid line denotes the flowback Bayesian update.

## M DERIVATION FROM EQ. 6 TO EQ. 9

Here we provide a formal derivation of the Bayesian Flow Distribution (Eq. 9) which is essentially the Eq. 172 to Eq. 185 in BFN (Graves et al., 2023). Firstly, we consider marginalizing the y from the sender distribution with the Bayesian update function in Eq. 6 which results in a distribution over $\theta$, we term it as Bayesian update distribution:

$$p_U\left(\boldsymbol{\theta} \mid \boldsymbol{\theta}_{i-1}, \mathbf{x}; \alpha\right) = \underset{\mathcal{N}(\mathbf{y}\mid\alpha(K\mathbf{e}_\mathbf{x}-\mathbf{1}),\alpha K\boldsymbol{I})}{\mathbb{E}} \delta\left(\boldsymbol{\theta} - \frac{e^{\mathbf{y}}\boldsymbol{\theta}_{i-1}}{\sum_{k=1}^{K} e^{\mathbf{y}_k}\left(\boldsymbol{\theta}_{i-1}\right)_k}\right). \tag{25}$$

Then we derive the additive accuracy property with two consecutive Bayesian updates of noisy samples $y_a \sim \mathcal{N}\left(\alpha_a\left(K\mathbf{e}_x-\mathbf{1}\right), \alpha_a K\boldsymbol{I}\right)$ and $y_b \sim \mathcal{N}\left(\alpha_b\left(K\mathbf{e}_x-\mathbf{1}\right), \alpha_b K\boldsymbol{I}\right)$ as:

$$
\begin{aligned}
h\left(y_b, h\left(y_a, \theta_{i-2}\right)\right) &= \frac{\exp\left(y_b\right) \frac{\exp(y_a)\theta_{i-2}}{\sum_{k'=1}^{K}\exp\left((y_a)_{k'}\right)(\theta_{i-2})_{k'}}}{\sum_{k=1}^{K}\exp\left((y_b)_k\right)\frac{\exp((y_a)_k)(\theta_{i-2})_k}{\sum_{k'=1}^{K}\exp\left((y_a)_{k'}\right)(\theta_{i-2})_{k'}}} \\
&= \frac{\exp\left(y_b\right)\exp\left(y_a\right)\theta_{i-2}}{\sum_{k=1}^{K}\exp\left((y_b)_k\right)\exp\left((y_a)_k\right)\left(\theta_{i-2}\right)_k} \\
&= \frac{\exp\left(y_a+y_b\right)\theta_{i-2}}{\sum_{k=1}^{K}\exp\left((y_a+y_b)_k\right)\left(\theta_{i-2}\right)_k} \\
&= h\left(y_a+y_b, \theta_{i-2}\right).
\end{aligned}
\tag{26}
$$

With the property of Gaussian distribution, we have that:

$$y_a + y_b \sim \mathcal{N}\left(\left(\alpha_a+\alpha_b\right)\left(K\mathbf{e}_\mathbf{x}-\mathbf{1}\right), \left(\alpha_a+\alpha_b\right)K\boldsymbol{I}\right). \tag{27}$$

Then the two-step update could be actually merged into a one-step update of added accuracies, i.e.:

$$\underset{p_U(\boldsymbol{\theta}_{i-1}\mid\boldsymbol{\theta}_{i-2},\mathbf{x};\alpha_a)}{\mathbb{E}} p_U\left(\boldsymbol{\theta}_i \mid \boldsymbol{\theta}_{i-1}, \mathbf{x}; \alpha_b\right) = p_U\left(\boldsymbol{\theta}_i \mid \boldsymbol{\theta}_{i-2}, \mathbf{x}; \alpha_a+\alpha_b\right) \tag{28}$$

The the Bayesian flow distribution in Eq. 9:

$$p_F(\boldsymbol{\theta} \mid \mathbf{x}; t) =$$

$$\underset{p_U(\boldsymbol{\theta}_1\mid\boldsymbol{\theta}_0,\mathbf{x};\alpha_1)p_U(\boldsymbol{\theta}_2\mid\boldsymbol{\theta}_1,\mathbf{x};\alpha_2)}{\mathbb{E}} \cdots \underset{p_U(\boldsymbol{\theta}_{n-1}\mid\boldsymbol{\theta}_{n-2},\mathbf{x};\alpha_{n-1})}{\mathbb{E}} p_U\left(\boldsymbol{\theta}_n \mid \boldsymbol{\theta}_{n-1}, \mathbf{x}; \alpha_n\right) = p_U\left(\boldsymbol{\theta}_n \mid \boldsymbol{\theta}_0, \mathbf{x}; \sum_{i=1}^{n}\alpha_i\right)$$

$$\tag{29}$$

Note $\sum_{i=1}^{n} \alpha_i$ could be substituted as $\beta(t)$ as the integral of the $\alpha(t)$ from $0$ to $t$ in the continuous setting. Substituting Eq. 25 into Eq. 29, we could obtain that:

$$p_F(\boldsymbol{\theta} \mid \mathbf{x}; t) = \underset{\mathcal{N}(\mathbf{y}|\beta(t)(K\mathbf{e_x}-\mathbf{1}),\beta(t)K\boldsymbol{I})}{\mathbb{E}} \delta\left(\boldsymbol{\theta} - \frac{e^{\mathbf{y}}\boldsymbol{\theta}_0}{\sum_{k=1}^{K} e^{\mathbf{y}_k}(\boldsymbol{\theta}_0)_k}\right). \tag{30}$$

Changing the notation from $y, \theta$ to $\mathbf{E}_{i,j}^{y}$ and $\mathbf{E}_{i,j}^{\theta}$ we could finish the derivation.

## N  FURTHER DISCUSSION OVER ADAPTIVE FLOWBACK

The generative process of BFN all start with the same initial points, e.g., vector $\left[\frac{1}{k}, \cdots, \frac{1}{k}\right]$ for k-class discrete variable, and ends with various generated samples. Hence, the variance of the input distribution first grows and then decreases [2], which could be intuitively understood as first exploring the endpoint of the generative process and then finetuning towards that endpoint. Motivated by this fact, we suggest that when the generative endpoints are approximately determined, it could be more favorable to eliminate the extra variance in $\theta$ from sequential Bayesian Update and only focus on finetuning $\theta$ based on the Bayesian flow of the current step output prediction. The condition of $\left\|\phi\left(\mathbf{G}^{\theta_t}, t\right) - \phi\left(\mathbf{G}^{\theta_{t-1}}, t-1\right)\right\|^2 \geq \epsilon$ is to test whether the predicted outputs of two consecutive steps are close enough, i.e. whether the steps $t$ and steps $t-1$ are generated towards the same targets. We use this as an indicator for whether the generation process is in a variance increase (exploration) or a variance decrease(finetuning) state.

## O  CONTINUOUS-STATE APPROACHES OR DISCRETE-STATE APPROACHES FOR GRAPH GENERATION

There have been several works making notable progress in continuous state-based methods, specifically spectrum-based approaches such as (Martinkus et al., 2022; Luo et al., 2023). More recently, several discrete state-based approaches that directly model the discrete variable of edges/nodes, e.g. (Vignac et al., 2022; Jo et al., 2024), have shown that combining the continuous state (e.g., spectrum) as an extra feature, the discrete state approaches could further achieve better results.

There is the particularity of graph data, whose form is discrete and important property enjoys continuity. And current empirical evidence (Vignac et al., 2022; Jo et al., 2024) suggests it could be favorable to take both into consideration. The motivation for choosing BFN lies in the fact that it could actually model the discrete variable in continuous categorical parameter space, which perfectly fits the discrete form and continuous property of graph data.

Additionally, though GraphBFN lies in the category of discrete state approaches, we would like to clarify that there is still a lot of research in the field to be conducted to finally answer the discrete or continuous question.

## P  RELATIONSHIP TO SPECTRAL GRAPH THEORY

Here we hope to provide a more detailed discussion of the relationship of proposed methods to spectral graph theory. The framework aims to introduce a new generative method that forms in the discrete state space and takes the intrinsic continuity of the topological information into consideration. The spectral graph theory provides an effective tool for us to implement and analyze the proposed approach.

From the implementation perspective, we use the spectral features of the input to enhance the network prediction following (Vignac et al., 2022). During both training and sampling, the spectral features of the parameters of the graphs are computed and directly appended to the network as additional inputs.

Additionally, we use spectral theory to help us analyze one key reason behind our method's performance improvement compared to previous approaches: the smooth transformation of graph topology leads to enhanced sample quality. As demonstrated in the ablation study in Appendix F, we use clustered graphs as an example to show that, for both diffusion-based methods and GraphBFN, a smoother transition of the sample's spectral features—such as the spectral gap in the case of clustered graphs—during the generation process significantly improves sample quality.

## Q   VISUALIZATION OF GENERATED SAMPLES

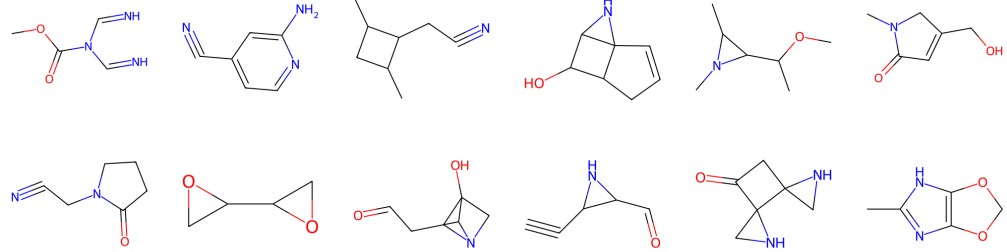

Figure 7: *Explicit* Hydrogens

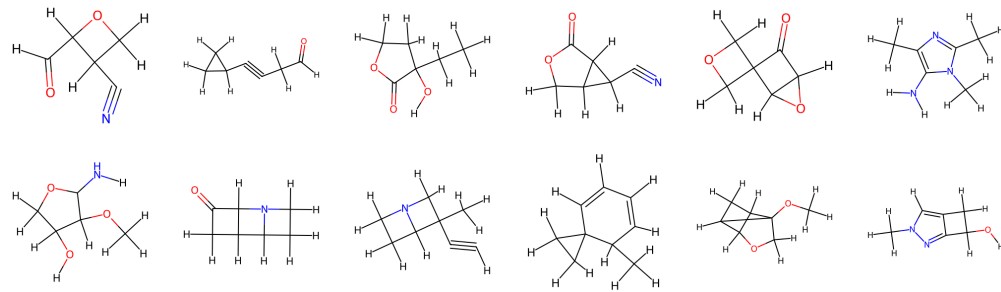

Figure 8: *Implicit* Hydrogens

Figure 9: Unfiltered samples generated by GraphBFN, trained on QM9

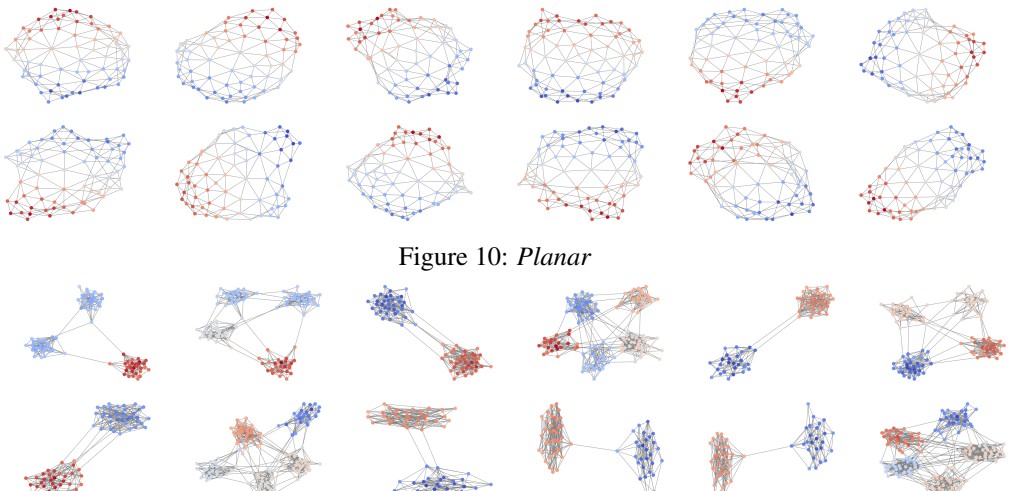

Figure 10: *Planar*

Figure 11: *SBM*

Figure 12: Unfiltered samples generated by GraphBFN, trained on abstract graph datasets

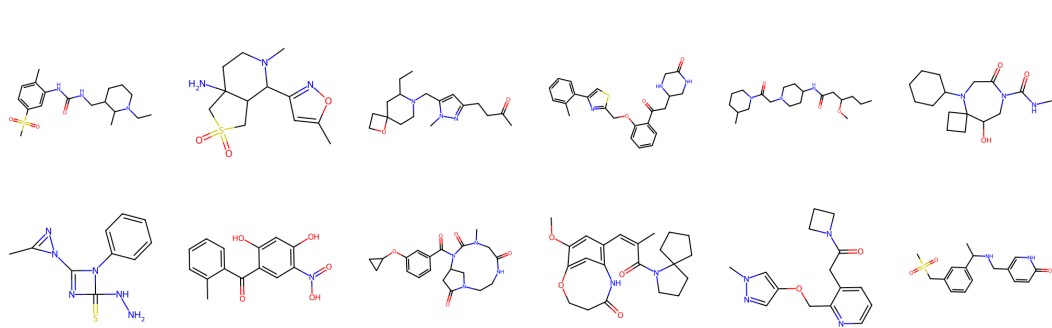

Figure 13: Unfiltered samples generated by GraphBFN, trained on ZINC250k with *implicit* hydrogens

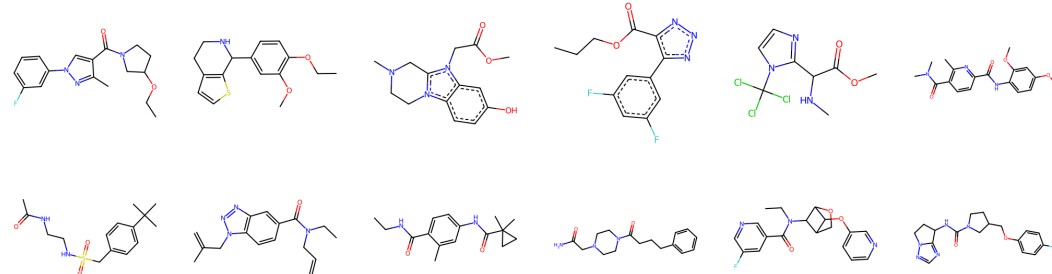

Figure 14: Unfiltered samples generated by GraphBFN, trained on MOSES graphs

