# OpenReview forum: "Smooth Probabilistic Interpolation Benefits Generative Modeling for Discrete Graphs"
_ICLR.cc/2025/Conference — Submitted to ICLR 2025_

### Official Review · Reviewer_MihX · 2024-10-31

**Soundness:** 3
**Presentation:** 2
**Contribution:** 3
**Rating:** 5
**Confidence:** 4

**Summary:**

This paper aims at the problem of graph generation. It models the graph generation process through Bayesian flow network. According to the topological information of the graph, a smooth probability interpolation method is defined to learn the topological properties of the graph.

**Strengths:**

1.Using BFN for graph generation is natural because it models the diffusion of discrete data very well.
The theoretical explanation and introduction of the article are very detailed, which strongly supports their views.

**Weaknesses:**

1.The authors do not clearly explain the relationship between spectral theory and their topological modeling.
The experiment selects too few datasets with different topological properties. The authors need to supplement common synthetic data such as community, ego, ba, etc., which have shown the efficient generation of their model on networks with various topological properties.

**Questions:**

1.Can the author give a more detailed explanation of the relation of spectral theory to your method?
Can the author give an the performance on other datasets?

**Details Of Ethics Concerns:**

No.

---

> ### Author Response · Authors · 2024-11-22
> **Response to Reviewer MihX**
>
> >**[W1 & Q1] Role of spectral theory in our approach + additional experiment results**
>
> Thank you for raising this important point. Below, we clarify the relation of spectral theory to our approach and present additional results on other common synthetic datasets.
>
> **Relationship to spectral theory.**  Our method employs the concept of spectral theory in two aspects: 1) model design and 2) the analysis of our method's performance gain.
>
> With respect to model design, the way we use the spectral features of graphs to enhance network prediction follows directly from previous methods (e.g. DiGress [1]). During both training and sampling, the spectral features of the intermediate graph entity are computed and directly appended to the network as additional inputs to aid its prediction. The only difference lies in that the GraphBFN computes the features over the parameter $\mathbf{G}^{\theta_t}$ of the probabilistic matrix $\mathbf{G}^{\theta}$ instead of the noisy graph.
>
> Additionally, we use spectral theory to help us analyze one key reason behind our method’s performance improvement compared to previous approaches: the smooth transformation of graph topology leads to enhanced sample quality. As demonstrated in the ablation study in Appendix F, we use clustered graphs as an example to show that, for both diffusion-based methods and GraphBFN, a smoother transition of the sample’s spectral features—such as the spectral gap in the case of clustered graphs—during the generation process significantly improves sample quality.
>
> **Additional experiment results.** In Appendix K.3, we presented our results on additional synthetic datasets, Ego-small, community-small, and Enzymes. GraphBFN's competitive capacity in modeling synthetic datasets remains, as we see that it shows superior performance on Community-small and competitive performance on Ego-small and Enzymes. We also note that for **Community-small** and **Ego-small**, the metrics are very saturated due to the limited complexity of datasets; as we can see, the SOTA results are very close to the training set's performance.
>
> [1] Vignac etal., 2022. "DiGress: Discrete Denoising diffusion for graph generation"
>
> ---
> We hope the above discussion could address your concerns.

---

> > ### Comment · Reviewer_MihX · 2024-11-25
> > **Response to Rebottal**
> >
> > Thanks for your rebottal. I hope that the author will be able to elaborate on the problem of spectral theory in subsequent editions. I will keep my score the same.

---

> > > ### Author Response · Authors · 2024-11-27
> > > **Thanks for your Response!**
> > >
> > > Thank you for taking the time to review our work. We have incorporated a discussion on the relationship between our proposed method and spectral theory in Appendix P. We kindly request your feedback on whether these revisions sufficiently address your concerns.
> > >
> > > Best regards,
> > >
> > > The Authors

---

### Official Review · Reviewer_dqky · 2024-11-03

**Soundness:** 2
**Presentation:** 3
**Contribution:** 3
**Rating:** 5
**Confidence:** 3

**Summary:**

This paper introduces Graph Bayesian Flow Networks (GraphBFN), a novel approach to graph generation using Bayesian Flow Networks (BFNs). Unlike traditional discrete diffusion models, GraphBFN uses a continuous latent variable, created by sampling infinitely from a categorical distribution, to smoothly interpolate between a prior state and the desired graph structure. By mapping this continuous latent to a probability simplex, GraphBFN produces a probabilistic adjacency matrix, representing the likelihood of various edge types, which allows for more nuanced and smooth graph generation. This framework enables efficient and accurate modeling of complex graph structures, supports diverse node and edge features, and offers substantial speed improvements over diffusion-based methods. Extensive experiments demonstrate GraphBFN’s strong performance and efficiency in generating realistic, topologically accurate graphs.

**Strengths:**

- The paper applies a new framework to graph generation, namely Bayesian Flow Networks.
- If the results are statistically significant (see weaknesses), the model exhibits good performance across all benchmarks.
- The empirical evaluations seem to suggest a better sampling efficiency than graph diffusion models.

**Weaknesses:**

- The concept of noisy channel is unusual in graph generation
- What is p_theta on l131 ? It's not defined anywhere.
- The definition of p_phi is unclear. What is y_i ? Reading section it seems that it can estimate both the density of y_t conditional to all previous states as well as the density of x given the sequence y_1:t in a diffusion model fashion. The role of this model is therefore hard to understand
- It took me some time to understand that your time axis goes from noise to data. Since you’re comparing to diffusion, which does the opposite, I think it is worth mentioning that somewhere.
- I find this sentence line 163 very unclear :
“For the discrete diffusion model (Vignac et al., 2022; Austin et al., 2021),
the Eq. 4 could be seen as one possible variant of the variational distribution q(yt|x) in Eq. 1, i.e. the extension of uniform transition diffusion (Austin et al., 2021) based on continuous-time Markov chain (CTMC) theory (Campbell et al., 2022).”
What do you mean by extension ? It seems that Austin et al. 2021 is based on Campbell 2022 …
- The introduction of noisy distribution is unclear. I would focus on clearly presenting the noisy distribution and provide a brief explanation of how this is achieved in appendix
- I think it would be worth introducing the sender and receiver distributions in section 2.2, to help the reader making the connection between section 3.2 and your optimization objective.
- The paragraph on Adaptive Flowback is poorly written, especially l 309-312
- You should provide error bars for all your experiments. Due to the limited size of the SPECTRE benchmark datasets, the metrics can exhibit high variance, casting doubt on the performance of some models such as Grum (cf e.g results in Siraudin et al. 2024)

**Questions:**

- Diffusion allows for direct model supervision, while you regress again y_t, which is a noisy quantity. This is downside compared to diffusion
- Where has the right hand side term of equation 3 gone in your loss ?
- What are the benefits of you method on graphs that do not exhibit community patterns such as SBM ?

---

> ### Author Response · Authors · 2024-11-22
> **Response to Reviewer dqky (Part 1)**
>
> We thank the reviewer for the detailed comments and constructive suggestions. We address all your concerns in the following:
> >**W1: The concept of noisy channels is unusual in graph generation.**
>
> Thanks a lot for your suggestion. The noisy channel in our draft actually refers to the sender distribution, which could produce a perturbed version of the clean data sample given a predefined noise-level parameter. We agreed that the term could make it challenging for a broader audience. We updated the noisy channel to noisy distribution.
>
> >**W2:  What is $p_{\theta}$ on l131 ? It's not defined anywhere.**
>
> Sorry for causing the confusion. There is indeed a typo, $p_\theta$ actually should be $p_\phi$ on L131. We have addressed the typo to eliminate misunderstanding.
>
> >**W3. The definition of $p_\phi$ is unclear. What is y_i? Reading section it seems that it can estimate both the density of y_t conditional to all previous states as well as the density of x given the sequence y_1:t in a diffusion model fashion. The role of this model is therefore hard to understand**
>
> We apologize for the confusion caused.  Due to the page limit, we tend to introduce the necessary mathematical formulation in the main text and actually provide a more detailed introduction of $p_\phi$ in Appendix A. The density for $p_\phi$ could be referred to in detail in Eq. 17.
>
> $y_i$ is a variable for noisy samples. Intuitively, $p_\phi$, also known as receiver distribution, is a noisy distribution with the accuracy level and formulation same as the sender distribution $q(y_i | x)$; While the clean sample $x$ in the sender distribution is replaced by the prediction from the neural network conditioned on the noisy samples $(y_1,\cdots,y_{i-1})$, i.e. $\phi(\theta_i=f(y_1,\cdots,y_{i-1}))$.
> As the reviewer noticed, the Eq. 3 also involved a term of $p_\phi(x \mid \theta_n)$(Now updated as $p_O(x|\theta_n,\phi)$.)
> And there is actually a slight abuse of the notation, by unifying both the output distribution and receiver distribution implied by neural network as $p_\phi$. We agree that this could cause misunderstanding and have revised the corresponding parts (Eq.3) in the main text of the revised draft as suggested. We also suggest referring to Appendix A, especially Eq. 18 for better understanding.
>
> >**W4: It took me some time to understand that your time axis goes from noise to data. Since you’re comparing to diffusion, which does the opposite, I think it is worth mentioning that somewhere.**
>
> Thanks a lot for the helpful suggestion! We have added the explanation of the time axis in the revised draft to help better understand and compare the proposed framework.
>
> >**W5:  The sentence line in 163 is very unclear:**
>
> Sorry for causing the confusion. Actually, the mentioned sequence tends to express that Eq.4, the noisy distribution defined for continuous timesteps, could be seen as the extension of the uniform transition, which is originally defined over discrete timesteps in D3PM[1]. Extending the discrete transition to continuous time relies on the continuous-time Markov chain (CTMC) theory[2]. We revised the corresponding parts as suggested to eliminate misunderstanding.
>
> [1] Austin et al., 2021. “Structured Denoising Diffusion Models in Discrete State-Spaces.”
> [2] Campbel et al., 2022 “A Continuous Time Framework for Discrete Denoising Models.”
>
> >**W6: The introduction of noisy distribution is unclear. I would focus on clearly presenting the noisy distribution and provide a brief explanation of how this is achieved in appendix.**
>
> Thanks for the suggestions! We agree that it would be favorable to present the noisy distribution clearly, and we revised the corresponding parts following your suggestions in the revised draft.
>
> >**W7: I think it would be worth introducing the sender and receiver distributions in section 2.2, to help the reader make the connection between section 3.2 and your optimization objective.**
>
> This is a great point!  We previously tended to introduce BFN conceptually with minimal notations in section 2.2. As the reviewer mentioned, we notice that it could indeed cause difficulties in connecting the optimization objective with the framework. We have moved the introduction of the sender and receiver, which are originally in the Appendix A, to the main text following suggestions.

---

> ### Author Response · Authors · 2024-11-22
> **Response to Reviewer dqky (Part 2)**
>
> >**W8: The paragraph on Adaptive Flowback is poorly written, especially l 309-312.**
>
> We rewrote the mentioned paragraph as suggested. And we briefly introduce the intuition of adaptive flowback in the following paragraph.
> The generative process of BFN all starts with the same initial points, e.g. vector $[1/k, \cdots, 1/k]$ for k-class discrete variable, and ends with various generated samples.  Hence, the variance of the input distribution first grows and then decreases [2], which could be intuitively understood as first exploring the endpoint of the generative process and then finetuning towards that endpoint. Motivated by this fact, we suggest that when the generative endpoints are approximately determined, it could be more favorable to eliminate the extra variance in $\theta$ from sequential Bayesian Update and only focus on finetuning $\theta$ based on the Bayesian flow of the current step output prediction.
>
> How the  Adaptive Flowback works: The condition of $\left\|\phi\left(\mathbf{G}^{\theta_t}, t\right)-\phi\left(\mathbf{G}^{\theta_{t-1}}, t-1\right)\right\|^2 \geq \epsilon$ is to test whether the predicted outputs of two consecutive steps are close enough, i.e. whether the steps $t$ and steps $t-1$ are generated towards the same targets.  We use this as an indicator for whether the generation process is in a variance increase (exploration) or a variance decrease(finetuning) state.
>
> We have also added this detailed explanation in Appendix N.
>
> >**W9:  Error bars for all experiments.**
> This is a very good suggestion from [1], pointing out the limitation of the current benchmark for synthetic graphs. We have also cited [1] in our revised paper.
>
> To conduct a more solid evaluation of the proposed methods, we re-report our experiment results on all the datasets with 95% confidence intervals appended.
>
> | Dataset | Deg. | Clus. | Orbit | Spec. | V.U.N. |
> |---|:-:|:-:|:-:|:-:|:-:|
> | Planar | $0.0005 \pm 0.0002$ | $0.0294 \pm 0.0006$ |$0.0002 \pm 0.0001$ | $0.0046 \pm 0.0003$ | $96.7 \pm 3.3$ |
> | SBM | $0.0005 \pm 0.0001$ | $0.0560 \pm 0.0012$ | $0.0370 \pm 0.0013$ | $ 0.0053 \pm 0.0005$ | $87.5 \pm 2.0$
>
> | Dataset | Valid | FCD | NSPDK  | Scarf.
> |---|:-:|:-:|:-:|:-:|
> | QM9 without H | $ 99.73 \pm 0.16 $ | $ 0.101 \pm 0.017 $ | $ 0.0002 \pm 0.0000 $ | $ 0.9386 \pm 0.0980 $ |
> | ZINC250k | $ 99.22 \pm 0.49 $ | $ 2.116 \pm 0.127 $ | $ 0.0013 \pm 0.0004 $ | $ 0.5304 \pm 0.0191 $ |
>
> | Dataset | Val | Unique | Novel | Filters | FCD SNN | SNN | Scaf. |
> |---|:-:|:-:|:-:|:-:|:-:|:-:|:-:|
> | MOSES | $88.5 \pm 2.9$ | $99.8 \pm 0.2$ | $89.0 \pm 3.3 $ | $98.3 \pm 0.8$ | $1.07 \pm 0.11$ | $0.59 \pm 0.07$ | $10.0 \pm 0.8$ |
>
> [1] Siraudin et al., 2024. "Cometh: A continuous-time discrete-state graph diffusion model."
>
> >**Q1: Diffusion allows direct model supervision, while you regress again y_t, which is a noisy quantity. This is downside compared to diffusion.**
>
> We would like to further compare Diffusion Models and BFNs to answer the question and eliminate the concerns.
>
> Firstly, **the objective of both diffusion and BFNs is minimizing the KL divergence between over two distributions of the noisy sample variable $y_t$.** With $t=T$ stands from the data, diffusion models minimizing the KL of $D_{\mathrm{KL}}(q(y_t \mid y_{t-1},x) | p_\phi(y_t|y_{t-1}))$ (Eq. 5 in original DDPM[1]) and BFN correspondingly minimizes $D_{\mathrm{KL}}(q(y_t \mid x) | p_\phi(y_t|\theta_{t-1}=f(y_1, \cdots, y_{t-1})))$.
>
> Secondly, for both models, the objectives could be reparameterized/transformed into regressions of clean data or the linear transformation of clean data. Practically, if the $q$ and $p_\phi$ both take the form of Gaussian distribution, the KL will turn to the MSE between the means of Gaussian distribution. The objectives could turn to regression towards the clean samples (BFN) or linear transformation of clean samples (Eq. 12 in DDPM [1]).
>
> [1] Ho et al., 2020. “Denoising Diffusion Probabilistic Models.”
>
> >**Q2:  Where has the right hand side term of equation 3 gone in your loss**
>
> Thanks for pointing it out. The right-hand side term of equation 3 is termed as reconstruction loss. The $\theta_n$ being very close to the ground truth $x$, makes the reconstruction loss term negligible in the total loss. Here we directly follow the vanilla BFN [1] and remove the term in optimization for efficiency. The above discussion is added to the updated version.
>
> [1] Graves, et al., 2023"Bayesian Flow Networks."
>
> >**Q3. What are the benefits of you method on graphs that do not exhibit community patterns such as SBM ?**
>
> We would like to clarify that our method's ability to promote smooth topological transformation generation applies to ALL categories of graphs. In our paper, we only intend to use the community graph datasets (e.g. Community-20 and SBM) as examples to illustrate the importance of smooth topology transformation, as presented in Appendix F.
>
> ---
> We hope the above discussion could address your concerns.

---

> > ### Comment · Reviewer_dqky · 2024-11-25
> > **Response to Rebuttal**
> >
> > Thanks for taking the time to write a detailed rebuttal. I think that your modifications have made the paper clearer than it used to be.
> >
> > Overall, even though your approach seems to yield competitive results, I feel that paper deserves an additional revision to improve clarity, especially concerning section 3.3. I have updated my score accordingly.

---

> > > ### Author Response · Authors · 2024-11-27
> > > **Thanks a lot for your response!**
> > >
> > > Thank you for taking the time to review our work!  We have revised section 3.3 following your suggestions in the updated draft. We kindly request your feedback on whether these revisions sufficiently address your concerns.
> > >
> > > Best regards,
> > > The Authors

---

### Official Review · Reviewer_TvTi · 2024-11-05

**Soundness:** 2
**Presentation:** 2
**Contribution:** 2
**Rating:** 5
**Confidence:** 3

**Summary:**

This paper proposes Graph Bayesian flow networks which is a Bayesian flow networks for graph generation. Motivated from the claim that graph topology can be captured in a continuous space, Graph Bayesian flow networks were designed. The method was validated by 2d molecular graph generation.

**Strengths:**

Extension of Bayesian flow networks to graph generation domain.

**Weaknesses:**

If, as claimed, the graph topology information, particularly the spectrum of graph-related matrices, can indeed be well represented in continuous space, it might be feasible to design a generative model in the frequency domain [1]. Given this, the choice of a Bayesian flow network is unclear.

[1] Luo, Tianze, Zhanfeng Mo, and Sinno Jialin Pan. "Fast graph generation via spectral diffusion." IEEE Transactions on Pattern Analysis and Machine Intelligence (2023).

While spectral features are mentioned, there is no detailed explanation of how they are utilized.

Comparisons with the most closely related works are lacking. There is no discussion of differences from GeoBFN [2] and MolCRAFT [3], nor is there a performance comparison with the 2D molecular graphs generated by those models.

[2] Song, Yuxuan, et al. "Unified generative modeling of 3d molecules via bayesian flow networks." arXiv preprint arXiv:2403.15441 (2024).

[3] Qu, Yanru, et al. "MolCRAFT: Structure-Based Drug Design in Continuous Parameter Space." arXiv preprint arXiv:2404.12141 (2024).

Overall, revisions are needed to improve the clarity and readability of the paper.

**Questions:**

Where are figures 1 and 6?

Could the authors explain the rationale behind the scheduling in Eq. 13, or is it a heuristic design?

Where are a detailed explanation of how spectral features were used and an assessment of their importance in generating 2D molecular graphs?

Compared to generated 2D molecular graphs from GeoBFN and MolCRAFT, what is the performance gap?

---

> ### Author Response · Authors · 2024-11-22
> **Response to Reviewer TvTi Part 1**
>
> We thank the reviewer for the insightful comments, and we address all your concerns in the following.
>
> >**W1: Towards the motivation over the choice of  Bayesian flow networks in Graph Topology Generation**
>
> Thanks a lot for pointing it out and introducing related literature[1] to us which we have incorporated as related work in the revised draft. Here, we would like to clarify our motivation behind the framework choice.
>
> The reviewer mentioned the "whether discrete state-based method or continuous state-based method" question, which sharply corresponds to an important open research question in graph generative modeling. There have been several works making notable progress in continuous state-based methods, specifically spectrum-based approaches such as [1,2]. More recently, several discrete state-based approaches that directly model the discrete variable of edges/nodes, e.g. [3,4], have shown that combining the
> continuous state (e.g., spectrum) as an extra feature, the discrete state approaches could further achieve better results.
>
> There is the particularity of graph data, whose form is discrete and important property enjoys continuity. And current empirical evidence [3,4] suggests it could be favorable to take both into consideration. The motivation for choosing BFN lies in the fact that it could actually model the discrete variable in continuous categorical parameter space, which perfectly fits the discrete form and continuous property of graph data.
>
> Additionally, though GraphBFN lies in the category of discrete state approaches, we would like to clarify that there is still a lot of research in the field to be conducted to finally answer the discrete or continuous question. We have updated the above discussion in Appendix O to illustrate the motivation of GraphBFN further.
>
> [1]Luo et al., 2024. "Fast graph generation via spectral diffusion."
>
> [2]Martinkus et al., 2022. "SPECTRE: Spectral Conditioning Helps to Overcome the Expressivity Limits of One-shot Graph Generators."
>
> [3] Vignac etal., 2022. "DiGress: Discrete Denoising diffusion for graph generation"
>
> [4] Jo et al. Graph Generation with Diffusion Mixture
>
> >**W2:  While spectral features are mentioned, there is no detailed explanation of how they are utilized.**
>
> Sorry for causing the confusion. The usage of the extra features, including structural and spectral features, directly follows the previous work [1], and the only difference lies in that the GraphBFN computes the features over the parameter of the probabilistic matrix $\mathbf{G}^{\theta}$ instead of the noisy graph. We have included an introduction of how the extra features are used in Appendix.K.1, where a more comprehensive explanation can be found in Appendix B of [1].
>
> Here we briefly discuss how it is used and implemented in neural networks. Each spectral feature or structural feature could be categorized into node-level features and graph-level features. Specifically, a node-level feature holds the shape of (batch_of_graphs, node_in_graph), such as an estimation of the biggest connected component (using the eigenvectors associated with eigenvalue 0) ; a graph-level feature holds the shape of (batch_of_graphs, 1). After computing the node-level features $F\_n$ and graph-level features $F\_g$ based on $V^{\theta}$ and $E^{\theta}$, we concat $V^{\theta}$ and all $F\_n$ as new node representations  $V\_{\text{feat.}}$  and concat all $F\_g$ as the finally  graph-level feature  $y$. Then the inputs for the graph transformers in GraphBFN are the triple of $(V\_{\text{feat.}}, E^{\theta},y )$  similarly to Digress [1]
>
> [1] Vignac etal., 2022. "DiGress: Discrete Denoising diffusion for graph generation"

---

> ### Author Response · Authors · 2024-11-22
> **Response to Reviewer TvTi Part 2**
>
> >**Q1: Where are figures 1 and 6?**
>
> Thanks for pointing it out. We notice that Figures 1 and 6, as well as the visualization illustrating Table 4, may not displayed correctly on certain PDF viewers, including PDF Expert.
>
> To address this, we have embedded these figures as PNG files instead of PDFs, which we found to improve compatibility across various PDF viewers. Additionally, we have highlighted the captions of the affected figures in blue for easy identification. **This updated version has been pushed!**
>
> >**Q2: Could the authors explain the rationale behind the scheduling in Eq. 13, or is it a heuristic design?**
>
> This is a great question. In short, our scheduling design is an improvement upon the original design in vanilla BFN [1] that is determined empirically based on the intuitive of linearly increased entropy. Our design attempts to change the non-informative prior to the original design to be informative.
>
> In Eq. 13, $\beta(t)$ is the accumulative (integral of $\alpha(t)$ from $0$ to $t$). The key intuition behind the design is that for informative prior, the initial step 0 already contains information of the ground truth sample. Then we set $\alpha(0)$ to a positive constant $a$. Besides, we also follow the heuristic of vanilla BFN[1] to use the square scheduler for accumulated accuracy $\beta$ to ensure the entropy of the input distribution approximately changes linearly along the timesteps. Eq.13 is a very simple form that satisfies both properties. We have revised the paper and added the discussions in Appendix K.1.
>
> [1] Graves, et al., 2023"Bayesian Flow Networks."
>
> >**Q3: Where are a detailed explanation of how spectral features were used and an assessment of their importance in generating 2D molecular graphs?**
>
> The detailed explanation of how spectral features were used could refer to our response to **W2**. To demonstrate its importance, we further conduct ablation studies on the spectral features of GraphBFN and DiGress. The results can be found in the following table. We see that extra feature conditioning benefits both methods.
>
> | | Val. (%) &uarr; | Uni. (%) &uarr; | A. Stab. (%) &uarr; | M. Stab. (%) &uarr; |
> |--------------|-------------|--------------|----------------|----------------|
> | Train. Set | 97.8 | 100 | 98.5 | 87.0 |
> | DiGress + no feature | 92.3±2.5 | 97.9±0.2 | 97.3±0.8 | 66.8±11.8 |
> | DiGress | 95.4±1.1 | 97.6±0.4 | 98.1±0.3 | 79.8±5.6 |
> | GraphBFN + no feature | 96.9±0.5 | **98.1**±0.2 | 99.1±0.2 | 90.5±0.4 |
> | GraphBFN |**99.2**±0.1 | 94.9±0.5 | **99.4**±0.0 | **94.7**±0.4 |
>
> ---
> We hope the above discussion could address your concerns.

---

> ### Author Response · Authors · 2024-11-26
> **Response to Reviewer TvTi Part 3**
>
> **W3 & Q4: Comparision and Discussion of the most related works GeoBFN and MolCraft**
>
> Thanks a lot for pointing it out. We agree that it is beneficial to explicitly discuss GraphBFN with GeoBFN and MolCraft to better understand the scope and difference of our proposed method.
>
> Both GeoBFN and MolCraft are 3D generative models based on BFN, which focus on modeling the 3D coordinates to reflect the topology of the generated molecules. Hence, **both approaches do not directly model the edge variable in their frameworks**. The edges between different nodes are implicitly inferred from the generated coordinates and types, with simple rule-based approaches or openbabel in these approaches. However, GraphBFN focuses on modeling the topology of the discrete graphs, which directly models the discrete edges variable and could extend to more general settings such as abstract graph generation where there are no 3D structures.
>
> Also, we note that MolCraft is the pocket-conditioned version of GeoBFN, which is not directly comparable with our approach.
> Nevertheless, we agree that a direct comparison with GeoBFN helps us understand the performance difference between the 2D relation-based and 3D geometry-based approaches. Accordingly, we further include GeoBFN as a baseline for the QM9 with explicit hydrogen task and present the updated table in the following.
>
> We find that GraphBFN consistently outperforms GeoBFN with a notable margin over the topological-related metrics such as validity and atom/molecule stability. This result is not surprising because GraphBFN explicitly models graph topologies, while GeoBFN infers them from generated atoms and 3D positions of these atoms.
>
> | | Validity (%) &uarr; | Uniqueness (%) &uarr; | Atom Stable(%) &uarr; | Molecule Stable (%) &uarr; |
> |--------------|-------------|--------------|----------------|----------------|
> | Train. Set | 97.8 | 100 | 98.5 | 87.0 |
> | GeoBFN | 93.4±0.2 | **98.3**±0.2 | 98.78±0.8 | 88.4±0.2 |
> | DiGress | 95.4±1.1 | 97.6±0.4 | 98.1±0.3 | 79.8±5.6 |
> | **GraphBFN (ours)** | **99.2**±0.1 | 94.9±0.5 | **99.4**±0.0 | **94.7**±0.4 |

---

> ### Author Response · Authors · 2024-11-28
> **Seeking feedbacks during the reviewer-author discussion period**
>
> Dear Reviewer TvTi,
>
> As the reviewer-author discussion period is nearing its end, we kindly remind you that we are eager to receive your feedback on our response.
>
> We have provided a detailed discussion of our motivation, added more specifics about the empirical implementation, included an ablation study on spectral features, and offered an additional comparison with the suggested 3D-based approaches to further support our claims. With the deadline approaching, we would appreciate knowing if our responses have adequately addressed your questions and concerns. If so, we hope you might consider a more positive rating, which would help our method reach a wider audience.
>
> Thank you once again for your time, and we sincerely look forward to your feedback!
>
> Best regards,
>
> The Authors

---

> ### Author Response · Authors · 2024-11-29
> **Gentle Reminder:  We Appreciate Your Feedback!**
>
> Dear Reviewer TvTi,
>
> Thank you once again for your valuable comments and insights.
>
> As the reviewer-author discussion period is approaching its conclusion on December 2nd, we want to remind you that we are eagerly awaiting your feedback on our response to your comments.
>
> In our response, we have provided a comprehensive rebuttal addressing your concerns, including comparisons with suggested works and discussions on motivation and technical details.
>
> If you have any further questions or need clarification on any aspect of our work, please do not hesitate to contact us.
>
> Thank you for your time and effort. We truly look forward to hearing from you!
>
> Best regards,
>
> The Authors

---

> > ### Comment · Reviewer_TvTi · 2024-12-01
> >
> > Thank you for the clarification and further experiments. I updated the score.

---

### Official Review · Reviewer_BZZM · 2024-11-12

**Soundness:** 3
**Presentation:** 2
**Contribution:** 3
**Rating:** 6
**Confidence:** 3

**Summary:**

This paper introduces a graph generative model based on a Bayesian Flow Network, where noisy graphs serve as latent variables in the data generation process. The model employs sender and receiver distributions to define the posterior and joint distributions of these latent variables, along with an aggregated variable to efficiently calculate the ELBO objective. Additionally, the paper introduces several technical innovations, whose impacts are evaluated through comprehensive ablation studies.

**Strengths:**

1. While the Bayesian Flow Network is an established framework, adapting it for graph generation and introducing technical improvements bring novelty to the paper.

2. The experiments are comprehensive, and the results are promising. The ablation study demonstrates the effectiveness of the "Adaptive Flow Back" technique.

**Weaknesses:**

1. While the paper presents an interesting and promising approach to graph generation, the clarity of some key technical details could be improved to enhance understanding:

- Loss function: Further clarification is needed on the statistical assumptions over $q(y_1,\cdots y_n\mid x)$ that allow the derivation of Eq 4 from Eq 3, as well as on whether $\log p(x \mid \theta_n)$ is incorporated in the final loss function (Eq 11).

- The generative model: Additional background on the definitions of $y$ and $\theta$ would be helpful. Are these definitions
modeling choices, or do they result from specific assumptions on the data generation process?

- The derivation of Eq 9 from Eq 6 should be elaborated at least in the appendix.

- Adaptive Flow Back: Although the paragraph provides a thorough introduction to the technique, the motivation behind it and how the mechanism achieves this goal are not immediately clear.

2. (minor) Typos and misuse of notations

- Eq 3: A negative sign should precede the KL term, and $\log p_{\phi}(x \mid \theta)$ should be $E_{q} \log p_{\phi}(x \mid \theta)$.

- Eq 4: the latent variable should use the notation $y_t$  instead of $z_t$ for consistency.

- Line 172: the categories should be defined with $\textbf{c} = [c_1, c_2, \cdots, c_K]$ to align with "$K$ categories".

- Line 272: "W the following notations" -> "With the following notations"

**Questions:**

Please address the concerns I raised in the "Weaknesses" section. I am open to revising my rating if these issues are satisfactorily addressed.

---

> ### Author Response · Authors · 2024-11-22
> **Response to Reviewer BZZM**
>
> **We thank the reviewer for the constructive comments and detailed suggestions. We address all your concerns in the following:**
> 1.  **Regarding Technical Details:**
>     - Loss function: Sorry for causing confusion. Here we clarify the definition of sender distribution $q(y\_1,\cdots, y\_n|x)$ in detail. The sender distribution is a non-parameterized variational posterior distribution of the corresponding latent variable model in analogy to the **forward process** in DDPM [1], which is handly designed for simplicity and flexibility. Here the sender distribution $q(y_1, \cdots, y_n|x)$ is defined to be factorized, i.e. $q(y\_1,\cdots, y\_n|x) = \prod\_{i=1}^{n}q(y\_i|x)$, which leads to the derivation of ELBO formulation in Eq.(3).  And here is a slight abuse of notation $q_{noise}$ in Eq. (4). $q_{noise}$ here is actually a component we defined for helping understand and derive the final exact formulation of $q(y_t|x)$ in Eq. (5). We have refined the notations here to improve readability. And $log p(x|\theta_n)$ generally contributes little to the EBLO objective, as shown in [2], as $\theta_n$ is already very close to the original sample based on the definition. Hence for efficiency, we do not incorporate the term following [2]. We add all the above-updated discussions to the revised draft.
>     - Generative Model: Thanks for the constructive suggestions. We provide a brief discussion here and leave a detailed explanation in Appendix A of the revised draft. The key belief or assumption of the data generation process behind Bayesian Flow Networks is that the information growth should be made as smoothly/continuously as possible [2]. To this end, the framework considers modeling the belief of the data, which could be empirically represented by the continuous parameter ($\theta$) of the distributions for all kinds of modalities. The variable $y$ refers to the noisy version of the samples, which is used to define the trajectory over $\theta$ for training and sampling with the Bayesian update rule. Intuitively, by observing different levels of noisy sample $y$ sequentially of the ground truth, the belief of the data could be gradually approached to the original data. Hence, the favored property of the distribution over $y$ is the close-formed Bayesian Update rule over $\theta$, such as Gaussian distribution, etc.
>
>     - The derivation from Eq. (6) to Eq. (9) has already been integrated into the Appendix M following your suggestions to make the presentation more comprehensive.
>
>     - Towards the Motivation of Adaptive Flowback:  The generative process of BFN all start with the same initial points, e.g., vector $[\frac{1}{k}, \cdots, \frac{1}{k}]$ for k-class discrete variable, and ends with various generated samples.  Hence, the variance of the input distribution first grows and then decreases [2], which could be intuitively understood as first exploring the endpoint of the generative process and then finetuning towards that endpoint. Motivated by this fact, we suggest that when the generative endpoints are approximately determined, it could be more favorable to eliminate the extra variance in $\theta$ from sequential Bayesian Update and only focus on finetuning $\theta$ based on the Bayesian flow of the current step output prediction.
>
>     - How the mechanism achieves this goal: The condition of $\left\|\phi\left(\mathbf{G}^{\theta_t}, t\right)-\phi\left(\mathbf{G}^{\theta_{t-1}}, t-1\right)\right\|^2 \geq \epsilon$ is to test whether the predicted outputs of two consecutive steps are close enough, i.e. whether the steps $t$ and steps $t-1$ are generated towards the same targets. We use this as an indicator for whether the generation process is in a variance increase (exploration) or a variance decrease(finetuning) state.
>
> [1] Ho et al., 2020. “Denoising Diffusion Probabilistic Models.”
>
> [2] Graves, et al., 2023"Bayesian Flow Networks."
>
> 2. **Regarding Typos and misuse of notations**
>
> Thank you for the catches! We have corrected all these typos in the updated manuscript.
>
>
> ---
> Thanks again for your detailed suggestions. We have updated all the mentioned typos and notation issues as you suggested in the revised draft.

---

> > ### Comment · Reviewer_BZZM · 2024-11-28
> > **Thank you for clarifying my questions**
> >
> > I appreciate the authors' detailed responses to my concerns and their efforts to revise the manuscript for greater clarity. With the revisions made, I am inclined to recommend accepting the paper and have adjusted the score. I hope the authors incorporate their discussion on the motivation behind adaptive flow back and how the mechanism achieves its goals, as provided in the rebuttal, into the revised manuscript.

---

> > > ### Author Response · Authors · 2024-11-28
> > > **Thanks for your feedback！**
> > >
> > > We appreciate the reviewer for updating the score. The suggested revisions have been incorporated into Appendix N. If there's any additional information you require, please feel free to contact us. Thank you once again for your time and effort in reviewing our draft.

---

### Author Response · Authors · 2024-11-23
**A Possible Rendering Issue in Certain PDF Viewers**

We would like to bring to your attention a potential rendering issue with the initial version of our paper, as noted by Reviewer TvTi. Specifically, Figures 1 and 6, as well as the visualization illustrating Table 4, may not display correctly on some PDF viewers, including PDF Expert.

To address this, we have embedded these figures as PNG files instead of PDFs, which we found to improve compatibility across various viewers. Additionally, we have highlighted the captions of the affected figures in blue for easy identification in the updated version.

Please have a look!

Best,
Authors

---

### Meta-Review · Area_Chair_vvwp · 2024-12-20

**Metareview:**

The paper propose Graph Bayesian Flow Networks (GraphBFN) to improve on the graph generation problem. GraphBFN utilises the learning of graph features in a continuous latent space to build up the generation procedure for graphs that of discrete nature. Despite BFN being established framework, applying the framework to graph generation is an interesting approach to tackle the problem and the empirical and ablation studies shows improvement on various pre-specified graph statistics in settings including planar graph, stochastic block models (SBM), as well as 2D molecular generation. As multiple reviewers pointed out, the connection and necessity of graph spectral theory discussed in the paper is less clear and convincing, weakening the clarity and novelty of the manuscript. Given the interesting idea and improvement on empirical findings, the interesting work shall be recommended another round of revise and polish before getting accepted into a major conference.

**Additional Comments On Reviewer Discussion:**

Reviewers' concerns are partially addressed.

---

### Decision · Program_Chairs · 2025-01-22

Reject